# NeuroMixGDP: A Neural Collapse-Inspired Random Mixup for Private Data Release

Donghao Li[1], Yang Cao[1], Yuan Yao [1*]

[1]The Hong Kong University of Science and Technology

`dlibf@connect.ust.hk, ycaoau@connect.ust.hk, yuany@ust.hk`

Privacy-preserving data release algorithms have gained increasing attention for their ability to protect user privacy while enabling downstream machine learning tasks. However, the utility of current popular algorithms is not always satisfactory. Mixup of raw data provides a new way of data augmentation, which can help improve utility. However, its performance drastically deteriorates when differential privacy (DP) noise is added. To address this issue, this paper draws inspiration from the recently observed Neural Collapse (NC) phenomenon, which states that the last layer features of a neural network concentrate on the vertices of a simplex as Equiangular Tight Frame (ETF). We propose a scheme to mixup the Neural Collapse features to exploit the ETF simplex structure and release noisy mixed features to enhance the utility of the released data. By using Gaussian Differential Privacy (GDP), we obtain an asymptotic rate for the optimal mixup degree. To further enhance the utility and address the label collapse issue when the mixup degree is large, we propose a Hierarchical sampling method to stratify the mixup samples on a small number of classes. This method remarkably improves utility when the number of classes is large. Extensive experiments demonstrate the effectiveness of our proposed method in protecting against attacks and improving utility. In particular, our approach shows significantly improved utility compared to directly training classification networks with DPSGD on CIFAR100 and MiniImagenet datasets, highlighting the benefits of using privacy-preserving data release. We release reproducible code in `https://github.com/Lidonghao1996/NeuroMixGDP`.

## 1. Introduction

Private data publishing is a technique that involves releasing a modified dataset to preserve user privacy while enabling downstream machine learning tasks. While many private data publishing algorithms exist, traditional algorithms (e.g., DPPro [1], PrivBayes [2], etc.) based on releasing tabular data are not suitable for modern machine learning tasks involving complex structures such as images, videos, and texts. To tackle this, a series of deep learning algorithms have emerged, such as DP-GAN [3] and PATE-GAN [4], which are based on training a Deep Generative Model (DGM) to generate data with complex structures, such as images, texts, and audios. These methods generate fake data based on the trained DGM and publish it instead of the raw data to respect users' privacy. However, as empirically observed by Takagi et al. [5], these DGM-based methods often suffer from training instability, such as mode collapse and high computational costs and lead to low utility, which is defined as the usefulness of the private data. For example, in the case of classification datasets, utility can be measured by classification accuracy.

DPMix — a new data publishing technique proposed by Lee et al. [6] — does not rely on training deep generative models and has the potential to improve utility. DPMix, as opposed to DGM-based methods, directly adds noise to the raw dataset — thereby taking into account users' privacy — and publishes the noisy version of the dataset. Concretely, inspired by Zhang et al. [7], DPMix first mixes the data points by averaging groups of raw data (with group size $m$), then adds noise to each individual mixture of data points to respect privacy concerns, and finally publishes the noisy

---

*Correspondence to: Yuan Yao <yuany@ust.hk>.

mixtures. The influence of such an averaging operation on utility is of two-fold: on the one hand, the averaging reduces injected noise that can improve utility under a given DP budget; on the other hand, averaging over large groups of data may obscure data features that cause a loss of utility. Therefore, a trade-off is necessary for the mixup degree $m$ and Lee et al. [6] empirically observe a "sweet spot" on $m$ that maximizes the utility under a given privacy budget. However, characterizing the "sweet spot" of the mixup degree remains an open problem, even for simple linear models.

While DPMix [6] has the potential to improve upon existing DGM-based methods (e.g., [3]), its full potential is not realized due to a limitation in the input space. To be specific, the input space may suffer the broken "Non-Approximate Collinearity" (NAC) condition, a sufficient condition for mixup training to minimize the original empirical risk [8]. NAC requires that the Euclidean neighborhoods of samples in any class do not lie in the space spanned by samples from other classes. When NAC is not satisfied, the virtual samples generated through mixup are often close to raw samples with different labels, leading to label confusion in the training procedure. For example, such a failure is usually observed when there exist samples lying on the decision boundary, as their neighborhoods may directly include samples from other classes and therefore can be easily represented by samples from those classes.

In contrast to the defect observed in the input space, the recently discovered Neural Collapse (NC) phenomenon [9, 10] reveals a desirable property in the last layer feature space, which motivates us to explore mixup in this space. NC states that when the training loss reaches zero, the last layer features of neural networks collapse to their class mean, which is located at the vertices of a simplex Equiangular Tight Frame (ETF). Consequently, when NC occurs, the last layer features are maximally separated from each other. As a result, in the feature space, the failure of the NAC occurs much more rarely compared to that in the input space.

Furthermore, recent progress in foundational models has demonstrated the potential to create universal representations of data that can be utilized for various downstream tasks [11]. In real-world applications, publishing features is often regarded as a more privacy-preserving approach compared to sharing raw data. For example, in recent Kaggle competitions [12–16], anonymous features are shared for clinical and financial data. Therefore, in this paper, we propose a new mixup framework based on features of late layers in neural networks, leveraging the desirable properties of the NC phenomenon. ***Our main contributions in this framework lie in the following aspects.***

The original random weight mixup (RW-Mix) was proposed by Zhang et al. [7] as a data augmentation technique to improve utility without adding privacy protection noise. However, when differential privacy constraints are added, RW-Mix suffers from sensitivity blowup. To address this issue, we introduce arithmetic means on groups of samples, referred to as Avg-Mix, which outperforms various choices of random weight means, as discussed in Section 2.

Moreover, when the mixup degree is large, Avg-Mix with existing subsampling schemes like Poisson Sampling (PS) may experience another issue called **L**abel **C**ollapse (**LC**), where the mixed labels are concentrated on the average label, resulting in a loss of class identifiability after noise injection. To address this issue, we propose a novel **H**ierarchical **S**ampling (**HS**) technique in Section 2. HS selects and averages samples within a random subset of classes instead of all classes, which helps distinguish different classes and mitigates LC.

Last but not least, we introduce an asymptotically optimal mixup degree rate using GDP for our mixups. This allows us to optimize the mixup degree by minimizing the prediction error in linear models, thereby addressing an open problem in the literature, as discussed in Section 3.

In summary, we shall call our algorithm as **NeuroMixGDP**, for the basic Avg-Mix on neural features, and **NeuroMixGDP-HS** for the one incorporating Hierarchical Sampling. Through extensive experiments in Section 4, our framework has demonstrated a state-of-the-art privacy-utility trade-off compared to existing algorithms in feature publishing. Additionally, NeuroMixGDP-HS has shown superior performance to DPSGD on both CIFAR100 and MiniImagenet datasets, particularly in low-budget scenarios. This observation underscores the potential for data publishing algorithms to surpass task-specific algorithms and makes the publishing of neural features a valuable direction

in terms of practical application. Apart from tight DP guarantees, we also show NeuroMixGDP(-HS) could successfully defend against two attacks, namely model inversion attack and membership inference attack, as discussed in Section 5.

## 2. NeuroMixGDP: Average Mixup and Hierarchical Sampling

This section discusses two challenges that can arise when using the feature mixup framework: the sensitivity blowup of RW-Mix and label collapse. We examine how Avg-Mix and HS can be used to address these issues, respectively, and how these approaches informed the development of our novel designs, NeuroMixGDP(-HS).

**Reducing sensitivity via Avg-Mix.** First, we give a formal definition of sensitivity and DP mixup.

**Definition 2.1.** The $l_2$-sensitivity $\Delta_f$ of a function $f$ is defined with $S$ and $S'$, which are neighboring datasets meaning that they differ by only one element:

$$\Delta_f := \max_{S,S'} ||f(S) - f(S')||$$

**Definition 2.2.** Differentially Private Mixup: Given a index set $\mathbb{I}_t = \{i_1, ..., i_m\}$ with size $m$, mixup produces virtual feature-label vectors by linear interpolation:

$$\tilde{\boldsymbol{x}} = \sum_{j=1}^{m} \boldsymbol{w}_j \boldsymbol{x}_{i_j} + \mathcal{N}(0, (C_w C_x \sigma_x)^2 \boldsymbol{I}); \quad \tilde{\boldsymbol{y}} = \sum_{j=1}^{m} \boldsymbol{w}_j \boldsymbol{y}_{i_j} + \mathcal{N}(0, (C_w C_y \sigma_y)^2 \boldsymbol{I}),$$

where $\boldsymbol{w}_j = \min(C_w, \boldsymbol{w}'_j)$ for $1 \leq j \leq m$, $\boldsymbol{w}' \sim Dir(\boldsymbol{\alpha})$, $Dir$ represents the Dirichlet distribution, $\boldsymbol{\alpha}_j = \alpha$ for $1 \leq j \leq m$. $C_w, C_x, C_y$ are parameters for clipping and $\sigma_x$ and $\sigma_y$ control DP noise scale. When $m = 2$, the Definition 2.2 degenerates into two sample mixup by [7]. The hyper-parameter $\alpha$ controls the concentration of the weight vector $\boldsymbol{w}$: as $\alpha$ approaches 0, the virtual feature-target vectors are closer to one of the original samples, whereas as $\alpha$ approaches infinity, the virtual feature-target vectors are closer to the arithmetic mean of the original samples. We refer to mixup with $\alpha = \infty$ and $C_w = \frac{1}{m}$ as Avg-Mix, and for mixup with finite $\alpha$, we use the term Random Weight Mixup (RW-Mix). The choice of $\alpha$ controls the randomness of mixup weight that have a direct effect on the utility. On the one hand, RW-Mix with small $\alpha$, such as setting $\alpha \in [0.05, 0.5]$ as data augmentation [7], is widely known for improving the utility. On the other hand, Avg-Mix enjoys better utility when DP is considered as shown in Fig. 1. This can be explained by the strength of DP noise, which is proportional to the sensitivity of the mixup operation. The sensitivity of RW-Mix is 1 (or $C_w$ after weight clipping). The sensitivity of Avg-Mix is $1/m$, achieving the minimal for the convex combination of $m$ samples. Under DP contains, the strength of DP noise dominates the utility so we observe that Avg-Mix is always better than RW-Mix with a

---

**Algorithm 1** NeuroMixGDP

**Input:** Dataset $S$ with $n$ samples and $K$ classes, feature extractor $f_1(\theta_1, \cdot)$, mixup degree $m$, class subsampling rate $\boldsymbol{p}$, noise scale $\sigma_x$, $\sigma_y$, $l_2$ norm clipping bound $C_x, C_y$, output size $T$.
**Output:** DP feature dataset $S_{DP}$ for publishing.
**for** $i = 1$ **to** $n$ **do**
    Extract feature and clip: $\boldsymbol{x}_i = f_1(\theta_1, \boldsymbol{x}_i)$
    $\boldsymbol{x}_i = \boldsymbol{x}_i / \max(1, ||\boldsymbol{x}_i||_2 / C_x)$
    $\boldsymbol{y}_i = \boldsymbol{y}_i / \max(1, ||\boldsymbol{y}_i||_2 / C_y)$
**end for**
**for** $t = 1$ **to** $T$ **do**
    Generate $\mathbb{I}_t$ with ***Poisson subsampling.***
    **Avg-Mix and perturb:**
    $\tilde{\boldsymbol{x}}_t = \frac{1}{m} \sum_{i \in \mathbb{I}_t} \boldsymbol{x}_i + \mathcal{N}(0, (\frac{1}{m} C_x \sigma_x)^2 \boldsymbol{I})$
    $\tilde{\boldsymbol{y}}_t = \frac{1}{m} \sum_{i \in \mathbb{I}_t} \boldsymbol{y}_i + \mathcal{N}(0, (\frac{1}{m} C_y \sigma_y)^2 \boldsymbol{I})$
    $S_{DP} = S_{DP} \cup \{(\tilde{x}_t, \tilde{y}_t)\}$
**end for**

---

Procedure of ***Poisson sampling :***
**Input:** Mixup degree $m$
**Output:** Index set $\mathbb{I}_t$
$\boldsymbol{b}_i \sim Bernoulli(\frac{m}{n})$ for $i \in [n]$.
$\mathbb{I}_t = \{i | \boldsymbol{b}_i = 1, i \in [n]\}$.

---

**Procedure 2** Hierarchical sampling

**Input:** $m$, class subsampling rate $\boldsymbol{p}$.
**Output:** Index set $\mathbb{I}_t$ for mixup.
$\boldsymbol{a}_k \sim Bernoulli(\boldsymbol{p}_k)$ for $k \in [K]$.
$\mathbb{K}_t := \{k | \boldsymbol{a}_k = 1, k \in [K]\}$.
$\boldsymbol{q}_k = \frac{m}{n \boldsymbol{p}_k}$ for $k \in [K]$.
$\boldsymbol{b}_i \sim Bernoulli(\boldsymbol{q}_{y_i})$ for $i \in [n]$.
$\mathbb{I}_t = \{i | \boldsymbol{y}_i \in \mathbb{K}_t, \boldsymbol{b}_i = 1, i \in [n]\}$.

grid search on $C_w$. Based on this observation, we propose a new method called NeuroMixGDP, which performs Avg-Mix operation in the feature space. The algorithm is presented in Algorithm 1.

**Reducing Label Collapse via Hierarchical Sampling.** Fig. 2 Left displays the distribution of mixed labels for Avg-Mix with and without DP noise when $m = 1024$. When DP noise is not present, we observe that the mixed label generated by Avg-Mix with Poisson Sampling (PS) becomes more concentrated on the averaged label $\frac{1}{K}$, where $K$ is the number of classes, as the mixup degree increases, and no dominant label appears. We refer to this phenomenon as **L**abel **C**ollapse (LC), as the mixed label collapses to $\frac{1}{K}\vec{1}$. This can be explained by the central limit theorem (CLT), that $\tilde{y}$ converges to a Gaussian distribution whose variance decreases with the increase of $m$, when the sample size is large enough. Existing sampling schemes fail to avoid LC because they do not take the class into account during sampling.

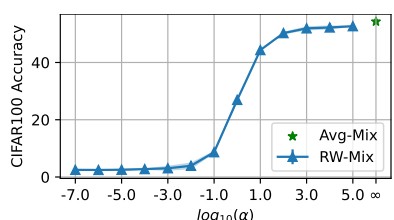

Figure 1: When DP noise is added, Avg-Mix consistently outperforms RW-Mix due to its lower sensitivity.

**Definition 2.3.** Poisson subsampling: Given a dataset $S$ of size $n$, the procedure Poisson subsampling outputs a subset of the data $\{S_i|a_i = 1, i \in [n]\}$ by sampling $a_i \sim Bernoulli(p)$ independently for $i = 1, ..., n$.

LC motivated us to develop a new mixup strategy that could generate strong class components while maintaining low sensitivity. Definition 2.4 describes the novel subsampling method.

**Definition 2.4.** Hierarchical Sampling $HiSample_{\boldsymbol{p},\boldsymbol{q}}$ : Given subsampling probabilities $\boldsymbol{p} \in [0, 1]^K, \boldsymbol{q} \in [0, 1]^K$ and a dataset $S = \{(\boldsymbol{x}_1, \boldsymbol{y}_1), ..., (\boldsymbol{x}_n, \boldsymbol{y}_n)\}$ with size $n$ and $K$ classes, the procedure HS first generates a subset of the label: $\mathbb{K} := \{k|\boldsymbol{a}_k = 1, k \in [K]\}$ by sampling $\boldsymbol{a}_k \sim Bernoulli(\boldsymbol{p}_k)$. Then outputs a subset of samples $\{(\boldsymbol{x}_i, \boldsymbol{y}_i)|\boldsymbol{y}_i \in \mathbb{K}, \boldsymbol{b}_{i,k=\boldsymbol{y}_i} = 1, i \in [n]\}$ by sampling $\boldsymbol{b}_{i,k} \sim Bernoulli(\boldsymbol{q}_k)$ independently.

As a result, we propose NeuroMixGDP-HS, which employs HS described in Procedure 2. HS randomly outputs samples from a subset of classes, so that only a few classes dominate in the mixed label. HS's subsampling law is provided in Proposition E.3. Fig. 2 shows that the mixed labels generated by Avg-Mix with HS have two distinct groups. One group concentrates on zero and the other around $1/3$. Since the sensitivity remains $\frac{1}{m}$, DP noise can be reduced as $m$ increases, allowing the two groups to remain distinguishable after noise injection.

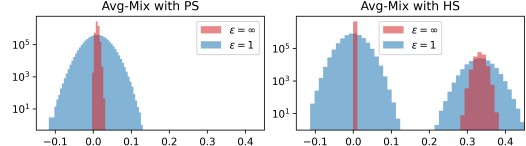

Figure 2: Histogram of $\tilde{\boldsymbol{y}}$ for PS and HS strategies. $\tilde{\boldsymbol{y}}$ from PS are concentrated on $\frac{1}{K}$ and overwhelmed by DP noise. HS produces strong class components, which are distinguishable from DP noises.

# 3. Gaussian Differential Privacy and Optimal Mixup Degree

In this section, we present the privacy guarantee of Algorithm 1 within the framework of GDP. Specifically, we provide an asymptotically optimal rate of mixup degree $m = \nu n/\sqrt{T}$, where the constant $\nu$ could be further chosen to maximize the utility of Algorithm 1 in linear models.

## 3.1. Privacy Guarantee of Algorithm 1

We present the privacy guarantee of Algorithm 1 within GDP framework in Theorem 3.1 , which states that with a properly specified choice of $\sigma_x$ and $\sigma_y$, Algorithm 1 is asymptotic $\mu$-GDP when $m\sqrt{T}/n \to \nu$ for a constant $\nu > 0$ as $T \to \infty$. Since [17] show that $\mu$-GDP corresponds to a series of $(\epsilon, \delta)$-DP, we can transfer $\mu$-GDP to $(\epsilon, \delta)$-DP using Lemma D.3 if needed.

**Theorem 3.1.** *For some $\lambda, \mu > 0$, let $\sigma_x$ and $\sigma_y$ be specified as*

$$\sigma_x = \frac{\sqrt{\lambda^2 + 1}}{\lambda \sqrt{\ln\left(1 + \frac{\mu^2 n^2}{m^2 T}\right)}}, \sigma_y = \frac{\sqrt{\lambda^2 + 1}}{\sqrt{\ln\left(1 + \frac{\mu^2 n^2}{m^2 T}\right)}}.$$

*Then Algorithm 1 is asymptotically $\mu$-GDP if $m\sqrt{T}/n \to \nu$ for a constant $\nu > 0$ as $T \to \infty$.*

The parameter $\lambda$ is used to balance the noise between the feature and label. In practical applications, it is often preferred to choose $\lambda \in [0.2, 4]$. The proof of Theorem 3.1 is provided in Appendix E.1, and a discussion on the relationship between the $f$-DP framework [17] and GDP is provided in Appendix D. Additionally, by setting $q_k = \frac{m}{n p_k}$ for HS, Theorem 3.1 can also characterize the privacy guarantee of NeuroMixGDP-HS. The proof of this result is presented in Appendix E.2, which involves developing a subsampling rule for HS.

In Theorem 3.1, the privacy guarantee is ensured only when $m\sqrt{T}/n \to \nu$ for a constant $\nu > 0$. In Appendix E.3, we show through the following simple one-sample test example to illustrate that such a requirement is strictly necessary to achieve both statistical utility and $\mu$-GDP protection. When $q_0\sqrt{T} \to \infty$, the limiting hypothesis test is trivially distinguishable, and there will be no privacy protection. When $q_0\sqrt{T} \to 0$, the limiting hypothesis test is completely indistinguishable, which corresponds to the perfect protection of privacy. However, in this case, the information of any sample will be overwhelmed by the Gaussian noise, which makes the released dataset not desirable for data analysis.

## 3.2. Characterizing the Sweet Spot in Linear Models

Lee et al. [6] empirically observes that there exists a "sweet spot" choice of mixup degree $m$ which can maximize the utility of mixup algorithms. As a theoretical complement to the empirical observation, in this section, we further characterize such a "sweet spot" rigorously on the basic linear models. Specifically, we show that the $\ell_2$ error on the least square estimator of the released private data is minimized by $m^* = \nu^* n\sqrt{T}$, where $\nu^* = \arg\max_\nu \nu \log(1 + \mu^2/\nu^2)$.

**Problem Setup.** Consider linear regression model defined by

$$\boldsymbol{y} = \boldsymbol{X}\beta^* + \boldsymbol{\epsilon},$$

where $\boldsymbol{X} \in \mathbb{R}^{n \times p}$ is the design matrix satisfying $\lambda_{\min} < \text{eigen}\left(\boldsymbol{X}^T\boldsymbol{X}/n\right) < \lambda_{\max}$ for some $\lambda_{\min}, \lambda_{\max} > 0$, $\boldsymbol{\beta}^* \in \mathbb{R}^p$ is the true regression parameter, $\boldsymbol{\epsilon} \in \mathbb{R}^n$ is the i.i.d. mean zero Gaussian noise with variance $\sigma^2$ and $y \in \mathbb{R}^n$ is the response vector. Each column of $X$ and $Y$ are normalized to have zero mean before the mixup procedure.

Let $\tilde{\boldsymbol{X}}, \tilde{\boldsymbol{y}}$ be the dataset generated after mixing up the features and adding DP noise, i.e.

$$\tilde{\boldsymbol{X}} = \boldsymbol{M}\boldsymbol{X} + \boldsymbol{E}_X, \quad \tilde{\boldsymbol{y}} = \boldsymbol{M}\boldsymbol{y} + \boldsymbol{E}_Y,$$

where $\boldsymbol{M} \in \mathbb{R}^{T \times n}$ is the random mixup matrix. Elements of $\boldsymbol{M}$ are i.i.d. random variables, which take $\frac{1}{m}$ with probability $\frac{m}{n}$ and take 0 with probability $1 - \frac{m}{n}$. $\boldsymbol{E}_X \in \mathbb{R}^{T \times p}$, $\boldsymbol{E}_Y \in \mathbb{R}^{T \times 1}$ are the random DP-noise matrices with their elements being i.i.d. Gaussian random variables respectively as defined in Algorithm 1 and Theorem 3.1. Here, for simplicity, we denote $C_X = C_x\sqrt{\lambda^2 + 1}/\lambda$ and $C_Y = C_y\sqrt{\lambda^2 + 1}$.

**Main Result.** Our target is to give an analysis on the $\ell_2$ loss of the least square estimator for $\beta^*$ based on $\tilde{\boldsymbol{X}}, \tilde{\boldsymbol{y}}$. Consider the well-known least square estimator $\tilde{\beta}$ given by

$$\tilde{\boldsymbol{\beta}} = \left[\tilde{\boldsymbol{X}}^T\tilde{\boldsymbol{X}}\right]^{-1}\tilde{\boldsymbol{X}}^T\tilde{\boldsymbol{y}}.$$

The following theorem characterizes $\|\tilde{\boldsymbol{\beta}} - \boldsymbol{\beta}^*\|_2$, whose proof is in Appendix G.

**Theorem 3.2.** *Consider $n, m, T \to \infty$ with $\frac{n}{T} \to \alpha$ for some $0 \le \alpha < 1$ and $\frac{m^2 T}{n^2} \to \nu^2$ for $\mu$-GDP. Then there holds the following with probability one when $n \to \infty$:*

$$\|\tilde{\boldsymbol{\beta}} - \boldsymbol{\beta}^*\|_2 \le \underbrace{\frac{2C_X\|\boldsymbol{\beta}^*\|_2 + 2C_Y}{\sqrt{\lambda_{\min}}(1 - \sqrt{\alpha})} \cdot \frac{1}{\sqrt{\nu \ln\left(1 + \frac{\mu^2}{\nu^2}\right)}} \cdot \frac{T^{\frac{1}{4}}}{\sqrt{n}}}_{bias} + \underbrace{\frac{2\sigma(1 + \sqrt{\alpha})}{(1 - \sqrt{\alpha})}\sqrt{\frac{\lambda_{\max}}{\lambda_{\min}}} \cdot \sqrt{\frac{p}{n}}\ln n}_{variance}. \quad (1)$$

*Remark* 3.3. For fixed $p$ and $T \to \infty$, Eq. (1) suggests that the estimation error is dominated by the *bias*, where the bias is minimized by choosing $\nu^* = \arg\min_\nu \nu \ln\left(1 + \mu^2/\nu^2\right)$. Fig. 11 validates by simulations with settings in Appendix J.1 that Eq. equation 1 reflects the optimal choice of $\nu^*$ well. Furthermore, Theorem 3.2 also suggests that $\nu^*$ increases when $\mu$ increases, which is confirmed by both simulation experiments (Fig. 11) and real-world classification tasks (Fig. 3).

**Comparison with Theorem 2 in Lee et al. [6].** Lee et al. [6] provided a training MSE bound with the help of the convergence theory of SGD by taking $\mathbb{E}[\boldsymbol{M}^T\boldsymbol{M}] = \boldsymbol{I}/(mn)$ and $m = o(n)$, which ignores the random matrix effect of $\boldsymbol{M}$. Theorem 2 in Lee et al. [6] states that given $T$, the training MSE of a linear regression model is a monotonically increasing function of $\sigma_x^2, \sigma_y^2$ both of which are decreasing as $m$ increase. Based on this observation, they suggest choosing the maximal $m = n$ for utility maximization since $\sigma_x^2, \sigma_y^2$ are minimized when $m = n$. This contradicts the empirical studies, e.g., Fig. 3. This is because the analysis by Lee et al. [6] relies on the assumption $m \ll n$ and if it is not satisfied, the random matrix effect of $\boldsymbol{M}$ can not be ignored. On the contrary, Theorem 3.2 here shows that the random matrix effect on the extreme eigenvalue distributions of $\boldsymbol{M}^T\boldsymbol{M}$ is necessary, under which the least square estimation error bound could be minimized at $m^* = \nu^* n/\sqrt{T}$.

# 4. Experiments

This section showcases the effectiveness of NeuroMixGDP(-HS) in improving utility and achieving state-of-the-art privacy-utility trade-offs among data-releasing algorithms. We also compare our method with DPSGD [18] and demonstrate comparable or better utility. Remarkably, in scenarios where the number of classes is large and the privacy budget is small, our method outperforms DPSGD by a significant margin, underscoring the potential of data-releasing algorithms.

**Experiment Setup**. Four widely used datasets are selected: MNIST [19], CIFAR10/100 [20], and Miniimagenet [21]. A detailed description of datasets will be shown in Appendix I.1. We set $\delta = 10^{-5} \approx 1/2n$ and show utility with different $\epsilon$ from $0.1$ to $10$. We report the mean and standard deviation of the accuracy from the last epoch over five independent experiments. The details of the experiment setting and models are listed in Appendix I.2. We also leave the discussion of the hyper-parameter setting in Appendix J.2.

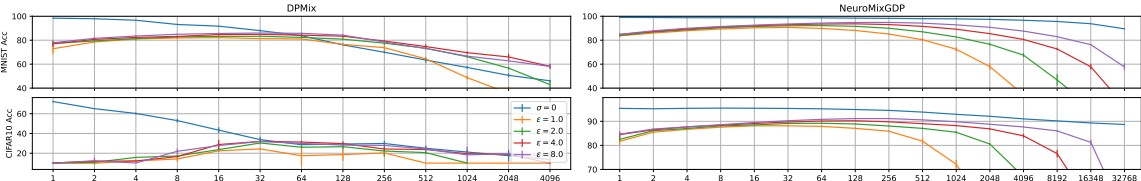

Figure 3: These figures show how utility changes w.r.t. $m$ and $\mu$ for DPMix and NeuroMixGDP.

**Improving Utility via Neural Collapsed Features.** Using NC features in NeuroMixGDP can potentially improve NAC condition and utility, especially for high mixup degree and large models.

First, as the feature extractor is not trained on the private data, exact collapse may not hold in the transfer learning setting. To investigate this issue, we evaluated the NC metric [9], which measures the magnitude of within-class covariance relative to between-class covariance of the features. Our findings indicate that feature extraction of a pre-trained ResNet model by SimCLR reduces the NC

metric from 45.34 and 164.09 to 0.49 and 3.03 for CIFAR10 and CIFAR100, respectively, suggesting that NC does occur in our transfer learning setting. To further validate if NC could induce NAC, we measure the Minimum Euclidean Distance (MED), as done by Chidambaram et al. [8]. A larger MED indicates a larger neighborhood satisfies NAC. The feature extraction increases MED values from 0.4003 and 0.3319 to 0.4874 and 0.4971 for CIFAR10 and CIFAR100 datasets, respectively (see Appendix I.5 for detailed experiment settings).

Next, in Fig. 3, we observe that as $m$ increases, the noise-free utility ($\sigma = 0$) of DPMix drops rapidly, which becomes the bottleneck for improving its utility. While the utility decay of NeuroMixGDP is not as severe as DPMix. The reason is that mixup can fail when NAC is not satisfied, especially when the mixing coefficient $\alpha$ is large [8]. The input spaces do not satisfied NAC and DPMix, corresponding to $\alpha = \infty$, is highly sensitive to NAC. However, by applying feature extractors, features with a simplex ETF structure are induced, where samples cannot be represented by an affine combination of samples from other classes, naturally satisfying NAC.

Table 1: CIFAR10 test accuracy (%) with different models. We show the convergence test accuracy before "+" and the improvement if we report the best accuracy during training after "+". Due to page limits, we leave results with $\epsilon = 1, 2, 4$ to Appendix Table 7.

|  | $\epsilon = 8$ | Non-private |
|---|---|---|
| DPMix CNN | 32.92+3.01 | 79.22 |
| DPMix ResNet-50 | 15.52+16.19 | 94.31 |
| DPMix ResNet-152 | 15.35+15.31 | 95.92 |
| Ours ResNet-50 | 80.75+0.04 | 94.31 |
| Ours ResNet-152 | 90.83+0.04 | 95.92 |

Table 1 summarizes that NeuroMixGDP could benefit from large feature extractors and significantly improve utility. In contrast, using larger pre-trained networks leads to a degradation of utility for DPMix. These phenomena suggest that the image space may not be an interpolation space in the sense that the mixup of input images does not necessarily lead to natural images with good features. Conversely, feature space favors interpolation, making linear classifiers work better. This is not surprising since feature extractors have a variety of invariant properties, such as ScatteringNets being (locally) translation/rotation/deformation invariant, and supervised or self-supervised pre-trained models being class or instance invariant. The interpolation of such invariant features naturally fits the interpolation of labels in linear models.

**Improving Utility via Hierarchical Sampling.** Fig. 4 investigates the effect of HS. Since CIFAR10 and CIFAR100 are balanced datasets, we set $\boldsymbol{p}_k = p$ for all $k$. Smaller values of $p$ correspond to mixup within fewer classes. When $p = 1.0$, HS reduces to PS. There are two key observations. First, HS consistently achieves better utility than PS and the optimal choice of $p$ depends on the privacy budget. Secondly, HS enjoys larger improvement on datasets with more classes, for example, CIFAR100. This can be explained by the fact that the mixed labels collapse to $1/K$. So, for datasets with larger $K$, mixed samples become closer to 0 and more indistinguishable after adding DP noise, resulting in greater degradation of utility. Therefore, by making mixed labels concentrate on $1/(Kp)$ and solving the label collapse problem, HS is more effective for datasets with large $K$.

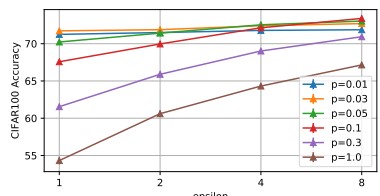

Figure 4: HS ($p < 1.0$) enjoys better utility than PS ($p = 1.0$).

**Utility Comparisons with Data Releasing Algorithm.** First, we present a comparison of 12 different methods reported in the literature on the MNIST test accuracy in Table 2, and the P3GM, DP-HP, and DP-MERF methods demonstrate good utility. Therefore, we test the utility of these three methods for feature release on the CIFAR10/100 dataset. However, since the DP-MEPF algorithm is not suitable for feature data release, we will use DP-MERF as a substitute as DP-MEPF is a variant of DP-MERF[26]. The privacy-utility trade-offs on the CIFAR10/100 dataset with and without feature extraction are reported in Table 3. For a fair comparison, we employ the same SimCLR feature extractor for all methods. Our observations reveal two important findings. First, data releasing without feature extraction fails to provide reasonable utility, particularly for the CIFAR100 dataset, where the test accuracies are close to random guessing. While feature release significantly enhances utility. Second, our method outperforms other methods in the feature releasing setting,

Table 3: Test accuracy (%) of data releasing algorithms with and without feature extraction.

| | CIFAR10 raw image releasing | | | | CIFAR10 SimCLR feature releasing | | | |
| --- | --- | --- | --- | --- | --- | --- | --- | --- |
| | P3GM | DP-MERF | DP-HP | DPMix | P3GM | DP-MERF | DP-HP | Ours |
| $\epsilon = 1$ | 17.47±3.27 | 16.08±0.96 | 16.63±1.63 | 24.36±1.25 | 23.25±2.37 | 59.21±5.35 | 40.16±2.64 | **90.46±0.15** |
| $\epsilon = 2$ | 15.00±2.82 | 26.13±0.73 | 15.35±0.53 | 30.37±1.30 | 29.22±2.92 | 63.68±13.32 | 82.88±1.45 | **91.19±0.16** |
| $\epsilon = 4$ | 14.86±6.37 | 28.22±0.76 | 15.32±0.09 | 31.97±0.71 | 38.39±3.21 | 65.62±9.94 | 86.37±3.90 | **91.88±0.12** |
| $\epsilon = 8$ | 18.20±4.16 | 28.68±0.82 | 13.88±0.38 | 32.92±0.94 | 44.02±6.29 | 67.48±10.48 | 87.17±3.08 | **92.53±0.06** |
| | CIFAR100 raw image releasing | | | | CIFAR100 SimCLR feature releasing | | | |
| | P3GM | DP-MERF | DP-HP | DPMix | P3GM | DP-MERF | DP-HP | Ours |
| $\epsilon = 1$ | 1.00±0.02 | 0.96±0.17 | 1.40±0.16 | 4.5±0.47 | 4.50±1.26 | 3.24±0.96 | 1.89±0.29 | **71.72±0.09** |
| $\epsilon = 2$ | 1.01±0.03 | 0.97±0.13 | 1.56±0.19 | 4.2±0.46 | 4.21±1.41 | 4.68±1.08 | 1.60±0.18 | **71.88±0.14** |
| $\epsilon = 4$ | 0.99±0.01 | 1.27±0.22 | 2.37±0.05 | 6.58±0.23 | 6.58±1.96 | 5.54±1.16 | 3.55±0.81 | **72.54±0.08** |
| $\epsilon = 8$ | 1.02±0.02 | 1.31±0.14 | 2.25±0.03 | 7.26±1.25 | 7.26±2.95 | 6.72±0.92 | 18.72±3.73 | **73.39±0.20** |

indicating its potential to become the new state-of-the-art approach. Furthermore, the advantage of our method is more pronounced in complex data distributions, such as CIFAR100. Unlike noisy DGMs used by other methods, which often struggle to capture the distribution due to insufficient training and low statistical utility, our method directly releases data without relying on such models.

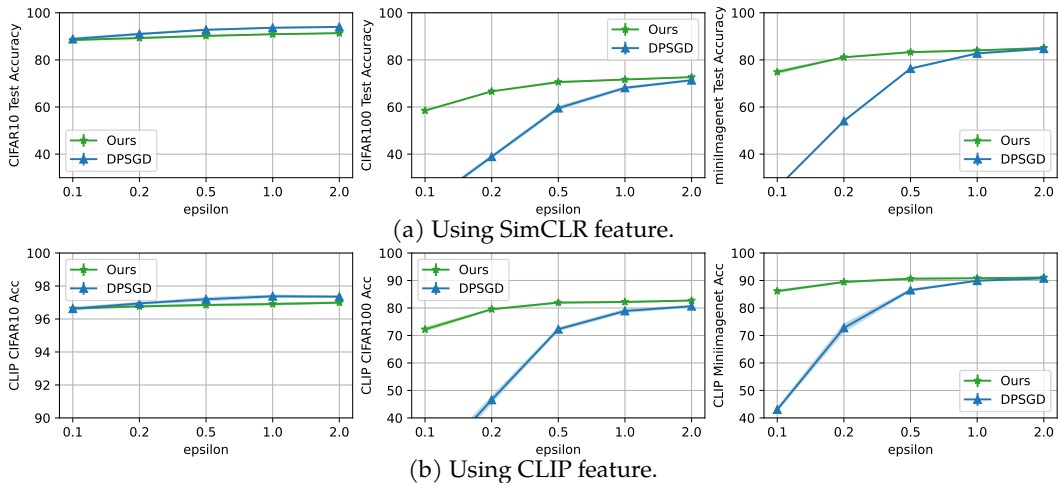

(a) Using SimCLR feature.

(b) Using CLIP feature.

Figure 5: Test accuracy of DPSGD and NeuroMixGDP-HS on CIFAR10/100 and Mini-Imagenet.

**Utility Comparisons with DPSGD.** To ensure a fair comparison, we use GDP accountant for both methods and conduct a grid search to determine the optimal hyper-parameters for DPSGD, as it is known to be sensitive to hyper-parameter [18]. Moreover, to demonstrate the potential of utilizing the multimodal foundation models, we also tried CLIP [11], as feature extractor, which has been shown that a fixed-weight feature extractor combined with a linear prob classifier surpassed the accuracy of end-to-end supervised training [11].

As shown in Fig. 5, NeuroMixGDP-HS, achieves comparable utility to DPSGD on the CIFAR10 dataset, and the gap between the two methods decreases as the privacy budget becomes tighter. When evaluating CIFAR100 and miniImagenet, which have more classes than CIFAR10, our method demonstrates a clear advantage, especially in the small-budget region. Besides, using CLIP feature further improves the utility, demonstrating the potential of utilizing foundational models.

Table 2: MNIST test accuracy reported in literature.

| Algorithm | $\epsilon = 1$ | $\epsilon = 10$ |
| --- | --- | --- |
| DP-GAN [3] | 40.36 | 80.11 |
| PATE-GAN [4] | 41.68 | 66.67 |
| GS-WGAN [22] | 14.32 | 80.75 |
| G-PATE [23] | 58.1 | 80.92 |
| DataLens [24] | 71.23 | 80.66 |
| DP-Sinkhorn [25] | - | 83.2 |
| DP-MERF [26] | - | 68 |
| PEARL [27] | 76.0 | 76.5 |
| P3GM [5] | 79.5 | 84.21 |
| DP-HP [28] | 82 | - |
| DP-MEPF [29] | 89.3 | 89.8 |
| DPMix [6] | 82.16 | 86.61 |
| **NeuroMixGDP** | **89.82** | **94.48** |
| **NeuroMixGDP-HS** | **92.71** | **96.17** |

The poor performance of DPSGD can be attributed to two factors. First, DP methods face a curse of dimensionality [30], as each dimension is injected with noise, and DPSGD has much larger noise dimensions. In the linear classifier setting, DPSGD injects noise to gradients with size $d_x \times K$, where $d_x$ is the dimension of the features and $K$ is the number of classes. In contrast, the noise dimension is $d_x + K$ for our method. Therefore, as $K$ increases, our method should achieve better utility than DPSGD. Furthermore, noisy gradients and limited iterations make it challenging for DPSGD to converge. In contrast, the optimization with privacy-preserving data released by our method can always obtain converged models without worrying about further leakage.

## 5. Attacking NeuroMixGDP

**Model Inversion Attack (MIA).** [31] has successfully recovered the raw data from released features. We want to know whether our method could defend against this type of attack, i.e. MIA. We modified the white-box attack in He et al. [31] to attack NeuroMixGDP. The details about the attack algorithm and more attacking results ($m = 8, 16, 32, 64$) are shown in Appendix K due to page limit. The visualization results are shown in Fig. 6.

The protection of NeuroMixGDP is three-fold. First, MIA on very deep models is weak, for example, Appendix Fig. 15 shows inverting ResNet-152 is quite hard in practice. However, this protection is weak if feature extractors are small and shallow. For example, when $m = 1$ and $\sigma = 0$, MIA could perfectly recover the raw images from the ScatteringNet features. Second, mixup makes the feature contains information from different images. When $m$ is large enough, for example, $m = 64$, the recovered images look entangled, and individual information is hidden.

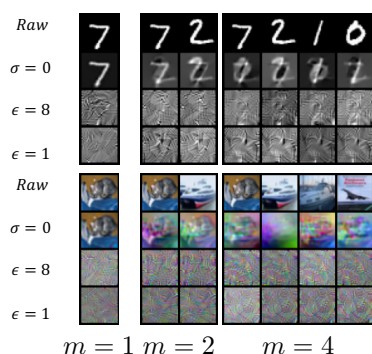

Figure 6: MIA against NeuroMixGDP.

However, when $m \le 4$, we can still recognize digits and objects from the entangled images. Finally, DP gives us a unified trade-off between $m$ and the noise scale and ensures the noise is strong enough to perturb any individual feature. For $\epsilon = 1$ or $8$, the recovered images look like pure noise with some random pattern, which might be caused by the Gaussian noise and total variation prior. This protection can be reliable even when the privacy budget is quite loose in theory.

**Membership inference on the CIFAR100 dataset.** We follow Yeom et al. [32], Salem et al. [33] and focus on algorithm-independent metrics: instance loss-based AUC. AUC without membership leakage should be 0.5. The AUC of the non-private baseline is significantly larger than 0.5, suggesting severe overfitting and membership leakage. NeuroMixGDP-HS with $\sigma = 0$, which represents pure mixup with $m = 64$ without DP noise, reduces the AUC a lot, showing the membership protection from mixup. Finally, adding DP noise could further reduce privacy leakage. For example, the AUC of NeuroMixGDP-HS with $\epsilon = 8$, which offers very loose protection in theory, is very close to perfect protection.

|  | Test Acc | AUC |
|---|---|---|
| Non-private | 84.45±0.02 | 0.5735±0.0000 |
| $\sigma = 0$ | 74.57±0.04 | 0.5160±0.0002 |
| $\epsilon = 8$ | 73.39±0.20 | 0.5117±0.0001 |
| $\epsilon = 4$ | 72.54±0.08 | 0.5117±0.0004 |
| $\epsilon = 2$ | 71.88±0.14 | 0.5107±0.0003 |
| $\epsilon = 1$ | 71.72±0.09 | 0.5090±0.0003 |

Table 4: Membership inference on CIFAR100.

## 6. Conclusion

This paper introduces a novel private data-releasing framework, which leverage feature mixup with desirable simplex ETF structures to achieve a state-of-the-art privacy-utility trade-off. Notably, we demonstrate that the general-purpose algorithms can outperform task-specific DP algorithms, such as DPSGD, in transfer learning scenarios where feature extractors are learned from public datasets. We hope that our work will inspire further research and applications of privacy-preserving data publishing, not only in the feature space but also beyond deep generative models.

# 7. Acknowledgement

The authors gratefully acknowledge National Natural Science Foundation of China Research Grants Council Joint Research Scheme Grant N_HKUST63520, Hong Kong Research Grant Council (HKRGC) Grant 16308321, and ITF UIM390. This research made use of the computing resources of the X-GPU cluster supported by the HKRGC Collaborative Research Fund C6021-19EF. The authors would like to thank Feng Ruan for helpful discussions and comments.

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

# A. Related Works

## A.1. Differential Private Data Releasing Algorithms

$(\epsilon, \delta)$-differential privacy [34] is a widely accepted rigorous definition of privacy. Under the differential privacy framework, we want to prevent attackers that want to determine whether some user exists in the private dataset by querying the dataset.

*Definition* A.1. A randomized algorithm $M$ satisfies $(\epsilon, \delta)$-differential privacy if for any two neighboring datasets $S$, $S'$ and any event $E$,

$$P(M(S) \in E) \leq e^{\epsilon} P(M(S') \in E) + \delta$$

There are two appealing properties for DP. First, DP allows analysis privacy loss of composition of multiple DP mechanisms, for example, stochastic gradient descent. Second, DP is robust to postprocess, which means that any function based on the output of $(\epsilon, \delta)$-DP mechanism is still $(\epsilon, \delta)$-DP.

Releasing private data with DP guarantee [1–3, 35, 36] receive more attention from machine learning community since downstream analysis could be regarded as post-process and cause no further privacy leakage. Current works focus on releasing raw data, for example, tabular data [2, 35] or image data Takagi et al. [5], Vinaroz et al. [28]. However, recent advancements in foundational models have showcased the potential for constructing universal representations of data [11, 37]. Besides, releasing features is considered to be privacy-preserving in real-world applications [12–16]. In this paper, we explore the potential of releasing private features with mixup.

## A.2. Neural Collapse

Recently, Papyan et al. [9] revealed a phenomenon named Neural Collapse, which has the following four properties of the last-layer features and classifiers. Fig. 7 provides a visualization to better understand NC.

(NC1) The within-class variation of the last-layer features converges to 0, which means that these features collapse to their class means. (NC2) The class means collapse to the vertices of a simplex equiangular tight frame (ETF) up to scaling. (NC3) Up to scaling, the last-layer classifiers each collapse to the corresponding class means. (NC4) The network's decision rule collapses to simply choosing the class with the closest Euclidean distance between its class mean and the feature of the test example.

*Definition* A.2. A $K$-simplex ETF is a collection of points in $\mathbb{R}^p$ specified by the columns of the matrix

$$\mathbf{A}^* = \sqrt{\frac{K}{K-1}} \mathbf{P} \left( \boldsymbol{I}_K - 1\frac{1}{K} \mathbf{1}_K \mathbf{1}_K^T \right)$$

where $\mathbf{1}_K$ is the ones vector, and $\boldsymbol{P} \in \mathbb{R}^{p \times K} (p \leq K)$ is a partial orthogonal matrix such that $\boldsymbol{P}^T \boldsymbol{P} = \boldsymbol{I}_K$.

The NC phenomena has been shown in self-supervised deep neural networks [10] and neural networks trained with MSE-lossHan et al. [38]. Moreover, the NC properties extend beyond test sets and also encompass adversarial robustness [9] and transfer learning [39].

Recently, researchers have conducted numerous studies on the theory of Neural Collapse. [40] provided the first analysis of the global optimization landscape of NC, while [41] introduced a surrogate model called the unconstrained layer-peeled model (ULPM) to investigate NC. They discovered that when ULPM is combined with the cross-entropy loss, it exhibits a benign global landscape for its loss function. Studying the landscape with MSE loss, [42] demonstrated that neural collapse solutions are the only global minimizers. Furthermore, [43] demonstrated that the benign global landscape can be extended to include label smoothing and focal loss. In their work, [44] proposed a surrogate model known as the unconstrained features model, while [45] revealed that the features of the minimizers can have a more intricate structure compared to the cross-entropy case.

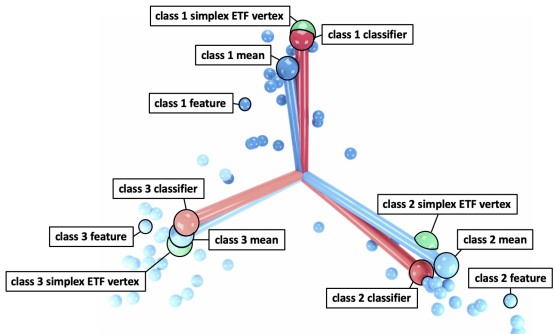

Figure 7: Diagram of NC from Han et al. [38].

## A.3. Mixup

Mixup, proposed by Zhang et al. [7], is a data augmentation technique that involves randomly linearly interpolating images and their corresponding one-hot labels. By encouraging neural networks to exhibit linear behavior, mixup enhances the generalization capabilities of different neural networks. In a study by Chidambaram et al. [8], they first examined a case where mixup failed and then provided a sufficient condition for mixup training to minimize the original empirical risk. They proved that NAC is a condition that enables mixup to achieve the same training error as empirical risk minimization training. This observation inspired the design of NeuroMixGDP, which utilizes neural feature extraction to induce NAC and enhances utility after the Avg-Mix process.

**Assumption A.3.** NAC: (Assumption 2.9. in Chidambaram et al. [8]) Consider $K$-class classification dataset where the supports $\mathbb{X}_1, ..., \mathbb{X}_K$ are finite sample sets of class $1, ..., K$. For any point $x \in \mathbb{X}_i$, there do not exist $u \in \mathbb{X}$ and $v \in \mathbb{X}_j$ for $j \neq i$ such that there is a $\lambda > 0$ for which $x = \lambda u + (1 - \lambda)v$.

# B. Limitation and broader impact

**Limitations:** This paper introduces NeuroMixGDP(-HS) and presents numerous experiments to showcase its improved utility. However, there are some limitations that need to be addressed in future research.

Firstly, NeuroMixGDP(-HS) relies on existing feature extractors. Nonetheless, recent advancements in deep learning have led to the development of powerful foundational models. These models "are trained on broad data (generally using self-supervision at scale) that can be adapted (e.g., fine-tuned) to a wide range of downstream tasks"[46]. Additionally, utilizing general-purpose feature extractors and constructing downstream tasks using private data has emerged as a new trend in private learning [18, 47]. In future work, we plan to explore the creation of feature extractors from private data, enabling the application of our method in a wider range of scenarios.

Secondly, this paper primarily focuses on computer vision classification tasks and lacks empirical results on other applications, such as classification in NLP datasets, where numerous pretrained models are also available. To address this, we aim to systematically investigate the performance of NeuroMixGDP(-HS) in other domains in future research.

**Broader impact:** This paper focuses on the release of private datasets for downstream tasks and demonstrates that constructing machine learning models on the released dataset for general purposes can yield better utility compared to training models directly on the private dataset using DPSGD in certain scenarios. This observation is expected to inspire further research on differential privacy data-releasing algorithms, exploring the potential benefits of leveraging released datasets for improved utility in various applications.

# C. Training with Noisy Mixup Labels

After getting the DP dataset, we can train a differentially private classifier on the DP features. One problem in building a classification model is the noisy mixup label. We solve this problem by clipping the negative values of $\bar{y}$ to 0 and then minimizing the generalized KL divergence for non-negative vectors $D(\boldsymbol{p}\|\boldsymbol{q})$, where $\boldsymbol{p}$ represents noisy mixup label and $\boldsymbol{q}$ represents the output of classifier after the Softmax layer.

*Definition* C.1. The generalized KL divergence for non-negative vectors $\boldsymbol{p}$, $\boldsymbol{q}$ is defined as:

$$D_{\mathrm{KL}}(\boldsymbol{p}\|\boldsymbol{q}) = \sum_i \left( \boldsymbol{p}_i \log \frac{\boldsymbol{p}_i}{\boldsymbol{q}_i} - \boldsymbol{p}_i + \boldsymbol{q}_i \right) \tag{2}$$

with the conventions $0/0 = 0$, $0 \log 0 = 0$ and $\boldsymbol{p}_i/0 = \infty$ for $\boldsymbol{p}_i > 0$.

# D. An Introduction to $f$-DP Framework

This section gives a self-contained introduction to $f$-DP framework. We will first introduce the definition of $f$-DP and $\mu$-GDP. Then show how $f$-DP handles subsampling and composition tightly. Since computing exact $f$-DP can be hard when $T$ is large, we introduce the "central limit theorem" phenomena and $\mu$-GDP approximation. To compare with other DP accountants, we also show the connection between $\mu$-GDP and $(\epsilon, \delta)$-DP.

## D.1. $f$-Differential Privacy

Under the differential privacy framework, we want to prevent an attacker who is well-informed about the dataset except for one single individual from knowing the presence or absence of the unknown individual. Wasserman and Zhou [48] first interpret such attack as a hypothesis testing problem:

$$H_0: \text{the true dataset is } S \quad \text{versus} \quad H_1: \text{the true dataset is } S'$$

We can defend against such an attack by injecting randomness to constrain the Type I error and Type II error of the hypothesis test over all possible rejection rules $\phi$. Let $P$ and $Q$ denote the distributions of $M(S)$ and $M(S')$ and Dong et al. [17] define the *trade-off function* between $P$ and $Q$ as:

$$T(P, Q) : [0, 1] \mapsto [0, 1]$$
$$\alpha \mapsto \inf_\phi \{1 - E_Q[\phi] : E_P[\phi] \le \alpha\},$$

where $E_P[\phi]$ and $1 - E_Q[\phi]$ are type I and type II errors of the rejection rule $\phi$. $T(P, Q)(\alpha)$ is thus the minimal type II error given type I error no more than $\alpha$. Proposition 2.2 in Dong et al. [17] shows that a function $f$ is a trade-off function if and only if $f$ is convex, continuous, non-increasing, and $f(x) \le 1 - x$ for $x \in [0, 1]$. Then $f$-DP is defined by bounding from the lower by $f$, the trade-off function between Type I and Type II error.

*Definition* D.1. [17] Let $f$ be a trade-off function. A (randomized) algorithm M is $f$-DP if:

$$T(M(S), M(S')) \ge f$$

for all neighboring datasets $S$ and $S'$.

A particular case of $f$-DP is the $\mu$-Gaussian Differential Privacy (GDP) [17], which is a single-parameter family of privacy definitions within the $f$-DP class.

*Definition* D.2. [17] A (randomized) algorithm $M$ is $\mu$-GDP if

$$T(M(S), M(S')) \geqslant G_\mu$$

for all neighboring datasets $S$ and $S'$, where $G_\mu(\alpha) = \Phi(\Phi^{-1}(1 - \alpha) - \mu)$ and $\Phi$ is the Gaussian cumulative distribution function.

Intuitively, GDP is to $f$-DP as normal random variables to general random variables. Let $G_\mu = T(\mathcal{N}(0,1), \mathcal{N}(\mu,1))$ be the trade-off function. Intuitively, GDP uses the difficulty of distinguishing $\mathcal{N}(0,1)$ and $\mathcal{N}(\mu,1)$ with only one sample to measure the difficulty of the above attack. Therefore larger $\mu$ means easier to distinguish whether a sample is in the dataset and larger privacy leakage. The most important property of GDP, as will be introduced below, is a central limit theorem that the limit of the composition of many "private" mechanisms converge to GDP.

To compare with other DP accountants, we can transform $\mu$-GDP to some $(\epsilon, \delta)$-DP using the following lemma by Dong et al. [17].

**Lemma D.3.** (Corollary 1 in Dong et al. [17]) *A mechanism is $\mu$-GDP if and only if it is $(\epsilon, \delta(\epsilon))$-DP for all $\epsilon \geqslant 0$, where*

$$\delta(\epsilon) = \Phi\left(-\frac{\epsilon}{\mu} + \frac{\mu}{2}\right) - e^\epsilon \Phi\left(-\frac{\epsilon}{\mu} - \frac{\mu}{2}\right).$$

## D.2. Subsampling with $f$-Differential Privacy

The subsampling technique will select a random subset of data to conduct private data analysis. A natural sampling scheme is PS, which selects each record with probability $p$ independently and outputs a subset with random size.

*Definition* D.4. (Poisson subsampling in Zhu and Wang [49]) Given a dataset $S$ of size $n$, the procedure Poisson subsampling outputs a subset of the data $\{S_i | a_i = 1, i \in [n]\}$ by sampling $a_i \sim Bernoulli(p)$ independently for $i = 1, ..., n$.

*Definition* D.5. (Uniform subsampling in Wang et al. [50]) Given a dataset $S$ of size $n$, the procedure Uniform subsampling outputs a random sample from the uniform distribution over all subsets of $S$ of size $m$.

Unlike Lee et al. [6], who choose Uniform sampling, we will choose the PS scheme in this paper for two reasons. First, compared with Uniform subsampling, PS scheme could enjoy a tighter privacy analysis. Theorem 6 in Zhu and Wang [49] shows a lower bound for $\epsilon$ under Uniform sampling scheme. While Wang et al. [50] prove such lower bound is also an upper bound for Gaussian mechanism when we switch to PS. That shows the Poisson sub-sampling scheme is at least as tight as Uniform sub-sampling. Second, Poisson subsampling under GDP has a closed-form expression for privacy budget and noise.

Next, we introduce how to deal with subsampling in $f$-DP framework and why the analysis is tight. Realizing that the un-selected data is not released and thus in perfect protection, Proposition A.1 in Bu et al. [51] states the trade-off function of sub-sampled mechanism satisfies the following property. Let $f_p := pf + (1-p)Id$ and $M \circ Sample_p$ denotes the subsampled mechanism, we have:

$$T(M \circ Sample_p(S), M \circ Sample_p(S')) \geq f_p$$

$$T(M \circ Sample_p(S'), M \circ Sample_p(S)) \geq f_p^{-1}.$$

The two inequalities imply the trade-off function is lower bounded by $\min\{f_p, f_p^{-1}\}$. The trade-off function of $M \circ Sample_p(S)$ should be $\min\{f_p, f_p^{-1}\}^{**}$, since $\min\{f_p, f_p^{-1}\}$ is not convex and can not be a trade-off function in general. Where $\min\{f_p, f_p^{-1}\}^{**}$ is the double conjugate of $\min\{f_p, f_p^{-1}\}$, which is the greatest convex lower bound of $\min\{f_p, f_p^{-1}\}$ and can not be improved in general. This suggests $f$-DP framework could handle subsampling tightly. Moreover, when $M$ is $(\epsilon, \delta)$-DP, Bu et al. [51] show that the above privacy bound strictly improves on the subsampling theorem in Li et al. [52].

## D.3. Composition with $f$-Differential Privacy

Another building block for DP analysis of the mixup-based method is composition. When analyzing iterated algorithms, composition laws often become the bottleneck. Fortunately, the $f$-DP

framework provides the exact privacy bound for the composed mechanism. Specifically, Theorem 4 in Dong et al. [17] states that composition is closed under the $f$-DP framework and the composition of $f$-DP offers the tightest bound. In contrast to $f$-DP, Dong et al. [17] show the exact privacy can not be captured by any pair of $\epsilon, \delta$. As for moment accountant, which is widely used for analysis of DPSGD, Theorem 1 and Theorem 2 in section 3.2 of [51] show that GDP offers an asymptotically sharper privacy analysis for DPSGD than MA in both $f$-DP and $(\epsilon, \delta)$-DP framework. Dong et al. [17], Bu et al. [51] also give a numerical comparison showing the advantage of GDP under the realistic setting of DPSGD.

## D.4. CLT Approximation with $\mu$-GDP

However, calculating the exact $f$ can be hard, especially when $T$ is large. Fortunately, there exists a central limit theorem phenomenon (Theorem 5 in Bu et al. [51]). Intuitively, if there are many "very private" mechanisms, which means the trade-off function is closed to $Id(\alpha) = 1 - \alpha$, the accumulative privacy leakage can be described as some $\mu$-GDP when $T$ is sufficiently large. Apart from asymptotic results, Theorem 5 in Dong et al. [17] shows a Berry-Esseen type CLT that gives non-asymptotic bounds on the CLT approximation.

# E. Privacy Analysis for NeuroMixGDP

## E.1. Proof for Theorem 3.1

Here we provide omitted proofs for Theorem 3.1. The following content consists of three parts. First, we need to consider one single step of Algorithm 1. Next, we use the refined composition theorem to bound the trade-off function of the composed mechanism. However, getting the exact solution is computationally expensive. Thanks to the central limits theorem of $f$-DP, we can approximate the trade-off function of the composed mechanism with GDP. We will also conduct simulation experiments to show that the $\mu$-GDP approximation can be quite precise when $T$ is finite in this section.

### E.1.1. Single-step of Algorithm 1

**Theorem E.1.** *Let $Sample_p(S)$ denotes the Poisson sampling process and $M$ be the averaging and perturbing process. Then the trade-off function of one step of Algorithm 1 $M \circ Sample_p$ satisfies:*

$$T(M \circ Sample_{\frac{m}{n}}(S), M \circ Sample_{\frac{m}{n}}(S')) \geq \frac{m}{n} G_{\sqrt{1/\sigma_x^2 + 1/\sigma_y^2}} + (1 - \frac{m}{n})Id$$

*Proof.* For simplicity, we now focus on a single step and drop the subscript $t$. Algorithm 1 releases both features and labels, and this process can be regarded as first releasing $\bar{x}$ (denoted as $M_x$) then releasing $\bar{y}$ (denoted as $M_y$). Therefore we can use the naive composition law of exact GDP (Corollary 2 in Dong et al. [17]) to get trade-off function $f$ for generated $(\bar{x}, \bar{y})$ (denoted as $M = M_x \circ My$).

First we consider $\bar{x}$, which has $l_2$-norm at most $C_x$ after clipping. Thus removing or adding one record in $S$ will change $\|\bar{x}\|_2$ by at most $\frac{C_x}{m}$, which means its sensitivity is $\frac{C_x}{m}$. So, the Gaussian mechanism $M$, which adds Gaussian noise $\mathcal{N}\left(0, (\frac{C\sigma_x}{m})^2\right)$ to each element of $(\bar{x}_t, \bar{y}_t)$ is $\frac{1}{\sigma_x}$-GDP by Theorem 1 of Dong et al. [17]. The same analysis could be applied to $M_y$. Then for $M$, we can apply naive composition law for exact GDP (Corollary 2 in Dong et al. [17]) and get it is $\sqrt{1/\sigma_x^2 + 1/\sigma_y^2}$-GDP. Finally, the trade-off function of $M \circ Sample_p$ can be described by Proposition A.1 in Bu et al. [51]. $\square$

### E.1.2. Composition

The refine composition law of $f$-DP (Theorem 4 in Bu et al. [51]) can be used directly for considering the composition in Algorithm 1. Here we could apply it and give proof for Theorem 3.1. Let $M'$ denotes Algorithm 1.

*Proof.* The size of the DP feature dataset is $T$, which means we conduct $T$ copies of mechanism $M \circ Sample_{\frac{m}{n}}$. Applying the above refined composition theorem, we have:

$$T(M'(S), M'(S')) \geq \left(\frac{m}{n}G_{\sqrt{1/\sigma_x^2 + 1/\sigma_y^2}} + (1 - \frac{m}{n})Id\right)^{\otimes T} := f$$

Moreover, if $S$ can be obtained by removing one record in $S'$, we will also have

$$T(M'(S), M'(S')) \geq f^{-1}$$

.

Combining the two inequalities, the trade-off function is lower bounded by $\min\{f, f^{-1}\}$. However, in general, $\min\{f, f^{-1}\}$ is not convex, thus not a trade-off function. We could use the double conjugate $\min\{f, f^{-1}\}^{**}$ as its trade-off function. Note that $\min\{f, f^{-1}\}^{**}$ is larger than $\min\{f, f^{-1}\}$ and can not be improved in general. $\qquad\square$

### E.1.3. CLT Approximation

Although we can have the exact trade-off function $\min\{f, f^{-1}\}^{**}$, it is hard to compute. Fortunately, there exists a central limit theorem phenomenon. The composition of many "very private" mechanisms can behave like some $\mu$-GDP (see Theorem 5 in Bu et al. [51]).

**Theorem E.2.** (*Theorem 5 in Bu et al. [51]*) *Suppose sample rate $p$ depends on $T$ and $p\sqrt{T} \to \nu$. Then we have the following uniform convergence as $T \to \infty$*

$$\left(pG_{1/\sigma} + (1 - p)\mathrm{Id}\right)^{\otimes T} = G_\mu$$

*where $\mu = \nu \cdot \sqrt{e^{1/\sigma^2} - 1}$ (Note that there is a typo in Bu et al. [51]).*

Here we could use Theorem 5 in Bu et al. [51] to prove Theorem 3.1.

*Proof.* Under the condition of Theorem 3.1, we can apply Theorem 5 in Bu et al. [51] directly. It gives:

$$\left(\frac{m}{n}G_{\sqrt{1/\sigma_x^2 + 1/\sigma_y^2}} + (1 - \frac{m}{n})Id\right)^{\otimes T} \to G_\mu$$

, where $\mu = \nu \cdot \sqrt{e^{1/\sigma_x^2 + 1/\sigma_y^2} - 1}$. Since $G_\mu$ is already a trade-off function, $\min\{G_\mu, G_\mu^{-1}\}^{**} = G_\mu$ and finish the proof of Theorem 3.1 $\qquad\square$

We also compare the asymptotic guarantee with exact numerical results with a typical setting of NeuroMixGDP and find the approximation is quite accurate even with $T \approx 200$. In Fig. 8, we numerically calculate the exact $f$-DP (red solid) given by Theorem 3.1 and compare it with its $\mu$-GDP approximation (blue dashed). We follow a typical setting of NeuroMixGDP we introduced in Appendix J.2. Specifically, $\mu = 0.5016$ (corresponding to $(2, 10^{-5})$-DP) and $\sigma = 0.8441$. In the original setting, $T = n = 50000$ is quite large. To best illustrate the fast convergence, we show the comparison of $T = n = 10, 50, 200$, respectively in Fig. 8. We also adjust $\frac{m}{n}$ to make all three cases satisfies $0.5016$-GDP. From the left figure, we can see that the two curves are getting closer when $T$ increases. The approximation $l_2$ and $l_\infty$ error shown in Fig. 9 also have the same trend. That suggests that when $T$ is sufficiently large, the CLT yields a very accurate approximation, which validates the Berry-Esseen type CLT (Theorem 5 in Dong et al. [17]). In practice, we often deal with very large $T$, for example, 60000 and 50000 for MNIST and CIFAR10, respectively and the CLT approximation should be quite accurate.

### E.2. Proof for Theorem 3 for NeuroMixGDP-HS

NeuroMixGDP-HS employs a different subsampling scheme so that different subsampling law should be used. The following proof show a novel subsampling law with its proof.

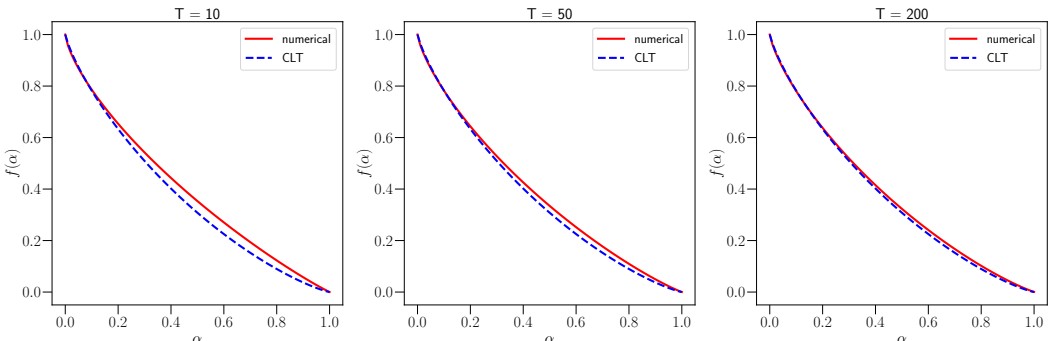

Figure 8: We compare the numerical results of $\left( \frac{m}{n} G_{\sqrt{1/\sigma_x^2+1/\sigma_y^2}} + (1 - \frac{m}{n})Id \right)^{\otimes T}$ with asymptotic GDP approximation. When $T$ increases, the two curves get closer to each other. We also show the approximation error w.r.t. $T$ in the following Fig. 9. That suggests the CLT approximation will be increasingly accurate as $T$ increase.

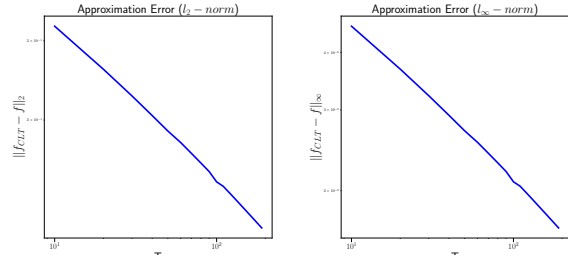

Figure 9: The figures shows the CLT approximation $\ell_\infty$ and $\ell_2$ error decrease when $T$ increase.

**Proposition E.3.** *if $M$ is $f$-DP, and $S' = S \cup \{x_0, y_0\}$, then:*

$$T\left(M \circ HiSample_{\boldsymbol{p},\boldsymbol{q}}(S), M \circ HiSample_{\boldsymbol{p},\boldsymbol{q}}(S')\right) \leq pf + (1-p)Id,$$

*where $\boldsymbol{q} \in [0,1]^K$, $\boldsymbol{q} \in [0,1]^K$, $p = \max_{k \in [K]} \boldsymbol{p}_k \boldsymbol{q}_k$.*

*Proof.* Without loss of generality, we can assume $S = \{(x_1, y_1), ..., (x_n, y_n)\}$ and $S' = \{(x_0, y_0 = k), (x_1, y_1), ..., (x_n, y_n)\}$. The outcome of the process $HiSample_{\boldsymbol{p},\boldsymbol{q}}$ when applied to $S$ is a bit string defined as $\boldsymbol{b} = (b_1, ..., b_n) \in \{0,1\}^n$. Bit $b_i$ decides whether $(x_i, y_i)$ is selected into the subset, denoted as $S_{\boldsymbol{b}}$. Let $\theta_{\boldsymbol{b}}$ be the probability that $\boldsymbol{b}$ appears. With this notation, $M \circ HiSample_{\boldsymbol{p},\boldsymbol{q}}(S)$ can be written as the following mixture:

$$M \circ HiSample_{\boldsymbol{p},\boldsymbol{q}}(S) = \sum_{\boldsymbol{b}} \theta_{\boldsymbol{b}} \circ M(S_{\boldsymbol{b}})$$

Similarly, $M \circ HiSample_{\boldsymbol{p},\boldsymbol{q}}(S')$ can also be written as a mixture, with an additional bit indicating the presence of $(x_0, y_0)$. Alternatively, we can divide the components into two groups: one with $(x_0, y_0)$ present, and the other with $(x_0, y_0)$ absent.

$$M \circ HiSample_{\boldsymbol{p},\boldsymbol{q}}(S) = \sum_{\boldsymbol{b}} \boldsymbol{p}_k \boldsymbol{q}_k \theta_{\boldsymbol{b}} \circ M(S_{\boldsymbol{b}} \cup \{x_0, y_0\}) + \sum_{\boldsymbol{b}} (1 - \boldsymbol{p}_k \boldsymbol{q}_k)\theta_{\boldsymbol{b}} \circ M(S_{\boldsymbol{b}})$$

Then applying Lemma A.2. of [51], for $k \in [K]$, we have:

$$T\left(M \circ HiSample_{\boldsymbol{p},\boldsymbol{q}}(S), M \circ HiSample_{\boldsymbol{p},\boldsymbol{q}}(S')\right) \leq \boldsymbol{p}_k \boldsymbol{q}_k f + (1 - \boldsymbol{p}_k \boldsymbol{q}_k)Id.$$

Finally, since [17] proved the monotonicity of subsampling operator, we can take the maximum over $k \in [K]$ and finish the proof. $\qquad \square$

Then applying the subsampling law of HS could finish the privacy analysis of NeuroMixGDP-HS. The HS is controlled by $p$ and $q$, and $\max_{k \in [K]} p_k q_k$ determines the largest probability of selecting one sample. NeuroMixGDP-HS set $q_k = \frac{m}{n p_k}$ for HS. Intuitively, this setting makes the probability of selecting every sample is equal to $\frac{m}{n}$, where $m$ is the mixup degree. Then replacing Proposition A.1 of [51] used in with Proposition E.3 will show that Theorem 3.1 also gives the GDP guarantee of NeuroMixGDP-HS.

## E.3. Hypothesis Testing interpretation of GDP

Consider the case that $T$ mixed samples are generated from $S = (x_1, x_2, \cdots, x_n)$ or $S' = (x_1, x_2, \cdots, x_n, x_{n+1})$. Let $H_0$ be the hypothesis that the dataset is $S$ and $H_1$ be the hypothesis that the dataset is $S'$. Let $\mathbf{p}_{i,j}$ are some independent Bernoulli random variables with parameter $q_0 = \frac{m}{n} \to 0$. The mixed samples $z_i$ for are generated by:

under $H_0$: $z_i = \sum_{j=1}^{n} \mathbf{p}_{i,j} x_j + \mathcal{N}(0, \sigma_z^2)$, under $H_1$: $z_i = \sum_{j=1}^{n} \mathbf{p}_{i,j} x_j + \mathbf{p}_{i,n+1} x_{n+1} + \mathcal{N}(0, \sigma_z^2)$.

Since DP offers the worst-case protection, it should necessarily offer protection for the case that $x_j = 0$ for $j \leq n$ and $x_{n+1} = 1$, where in this case : $z_i = q_i + \mathcal{N}(0, \sigma_z^2)$

under $H_0$: $q_i \overset{i.i.d.}{\sim} \text{Bernoulli}(q)$ and $q = 0$, under $H_1$: $q_i \overset{i.i.d.}{\sim} \text{Bernoulli}(q)$ and $q = q_0$.

As DP offers protection against any attack on membership, it should necessarily offer protection against the attack which conducts the hypothesis test through maximum likelihood estimators (MLEs). Proposition E.4 present the different asymptotic normality on the MLE of $q$ under $H_0$ and $H_1$. The proof of Proposition E.4 is in Appendix F.

**Proposition E.4.** *With $T \to \infty$ and $q_0 \to 0$, the distributions of the MLE $\hat{q}$ satisfy*

$$H_0: \sqrt{T} \hat{q} \left[ \exp\left(\frac{1}{\sigma_z^2}\right) - 1 \right]^{-1/2} \sim \mathcal{N}(0, 1), \qquad H_1: \sqrt{T} \hat{q} \left[ \exp\left(\frac{1}{\sigma_z^2}\right) - 1 \right]^{-1/2} \sim \mathcal{N}(A, 1),$$

*where $A := q_0 \sqrt{T} \left[ \exp\left(\frac{1}{\sigma_z^2}\right) - 1 \right]^{-1/2}$.*

By Proposition E.4, the hypothesis test based on $\hat{q}$ is equivalent to distinguishing two normal distributions with one observation. When $q_0 \sqrt{T} \to \infty$, $H_0$ v.s. $H_1$ is trivially distinguishable, and there will be no privacy protection. When $q_0 \sqrt{T} \to 0$, $H_0$ v.s. $H_1$ is completely indistinguishable, which corresponds to the perfect protection of privacy. However, in this case, the information of any sample will be overwhelmed by the Gaussian noise, which makes the released dataset not desirable for data analysis. Therefore, the condition $q_0 \sqrt{T} \to \nu$ ($\nu > 0$) in Theorem 3.1 is necessary to achieve both statistical utility and $\mu$-GDP protection.

## E.4. DP Guarantee with Renyi-DP

To compare Renyi-DP with the PS scheme, we also provide privacy bound for Algorithm 1 here. In experiments, we calculate the RDP privacy budgets using the autodp python package, which is officially released software of Zhu and Wang [49].

**Theorem E.5.** *Given $n, m, \sigma_x, \sigma_y, C_x, C_y, T$, Algorithm 1 satisfies $(\alpha, \epsilon_{RDP}(\alpha))$-Renyi-DP for integer $\alpha \geq 2$, where*

$$\epsilon_{RDP}(\alpha) = T \frac{1}{\alpha - 1} \log \left\{ (1 - \frac{m}{n})^{\alpha - 1} (\alpha \frac{m}{n} - \frac{m}{n} + 1) + \sum_{\ell=2}^{\alpha} \binom{\alpha}{\ell} (1 - \frac{m}{n})^{\alpha - \ell} \frac{m}{n}^{\ell} e^{(\ell-1)(\frac{\alpha}{2\sigma_x^2} + \frac{\alpha}{2\sigma_y^2})} \right\} \tag{3}$$

*Remark* E.6. We can convert RDP to $(\epsilon, \delta)$-DP by Proposition 3 in Mironov [53]. By minimizing over $\alpha$, we can get Algorithm 1 is $(\epsilon_{RDP}^*, \delta)$-DP:

$$\epsilon_{RDP}^* = \min_{\alpha \geq 2} \epsilon_{RDP}(\alpha) + \frac{\log(1/\delta)}{1 - \alpha}$$

*Proof.* As shown in Appendix G, one step of Algorithm 1 is composed of $M_x$ and $M_y$. For $M_x$, it satisfies $\epsilon_{M_x}(\alpha) = \frac{\alpha}{2\sigma_x^2}$ by Gaussian mechanism in Mironov [53]. For $M_y$, it satisfies $\epsilon_{M_y}(\alpha) = \frac{\alpha}{2\sigma_y^2}$. Therefore according to Proposition 1 of Mironov [53], $M$ satisfies:

$$\epsilon_M(\alpha) = \frac{\alpha}{2\sigma_x^2} + \frac{\alpha}{2\sigma_y^2}.$$

Then we consider the composition of subsampled mechanisms. According to Theorem 6 and Proposition 10 in Zhu and Wang [49], Gaussian mechanism applied on a subset from PS satisfies $(\alpha, \epsilon'_{RDP}(\alpha))$-Renyi-DP for any integer $\alpha \geq 2$, where,

$$\epsilon'_{RDP}(\alpha) = \frac{1}{\alpha-1} \log \left\{ (1 - \tfrac{m}{n})^{\alpha-1}(\alpha \tfrac{m}{n} - \tfrac{m}{n} + 1) + \sum_{\ell=2}^{\alpha} \left( \begin{array}{c} \alpha \\ \ell \end{array} \right) (1 - \tfrac{m}{n})^{\alpha-\ell} \tfrac{m}{n}^\ell e^{(\ell-1)\epsilon_M} \right\} \quad (4)$$

.

Then the overall privacy leakage can be calculated by the composition law of Renyi-DP (Proposition 1 in Mironov [53]). Therefore, Algorithm 1 satisfies $(\alpha, T\epsilon'_{RDP}(\alpha))$-RDP. $\qquad\square$

## F. Proof of Proposition E.4

*Proof.* First, we show that the MLE of $q$ are asymptotically normal as $T \to \infty$ under both $H_0$ and $H_1$. The log likelihood will be given by

$$\ell(q) = \sum_{i=1}^{T} \ln \left[ q \exp \left( -\frac{(z_i - 1)^2}{2\sigma_z^2} \right) + (1 - q) \exp \left( -\frac{z_i^2}{2\sigma_z^2} \right) \right], \quad (5)$$

$$= \sum_{i=1}^{T} \ln \left( 1 + q \left[ \exp \left( \frac{2z_i - 1}{2\sigma_z^2} \right) - 1 \right] \right) + \sum_{i=1}^{T} \ln \left[ \exp \left( -\frac{z_i^2}{2\sigma_z^2} \right) \right], \quad (6)$$

$$\overset{q \to 0}{\approx} q \sum_{i=1}^{T} \left[ \exp \left( \frac{2z_i - 1}{2\sigma_z^2} - 1 \right) \right] + \sum_{i=1}^{T} \ln \left[ \exp \left( -\frac{z_i^2}{2\sigma_z^2} \right) - 1 \right], \quad (7)$$

with the approximation

$$\frac{d\ell(q)}{dq} \overset{q \to 0}{\approx} \sum_{i=1}^{T} \left[ \exp \left( \frac{2z_i - 1}{2\sigma_z^2} \right) - 1 \right]. \quad (8)$$

The Fisher information matrix with a single observation $z_i$ is given by:

$$I_i = E \left[ \left( \frac{d\ell(q)}{dq} \right)^2 \right] \quad (9)$$

$$= \int \left( \exp \left( \frac{2z_i - 1}{2\sigma_z^2} \right) - 1 \right)^2 \frac{1}{\sigma_z \sqrt{2\pi}} \exp \left( -\frac{z_i^2}{2\sigma_z^2} \right) dz_i \quad (10)$$

$$= 1 + \int \frac{1}{\sigma_z \sqrt{2\pi}} \exp \left( \frac{-z_i^2 + 4z_i - 2}{2\sigma_z^2} \right) dz_i \quad (11)$$

$$- 2 \int \frac{1}{\sigma_z \sqrt{2\pi}} \exp \left( \frac{-z_i^2 + 2z_i - 1}{2\sigma_z^2} \right) dz_i \quad (12)$$

$$= \left[ \exp \left( \frac{1}{\sigma_z^2} \right) - 1 \right] \quad (13)$$

$$(14)$$

Then with $q_0 \to 0$, the maximum likelihood estimator $\hat{q}$ will be asymptotically

1. Under $H_0$, $\sqrt{T}\hat{q} \sim \mathcal{N}\left(0, \left[\exp\left(\frac{1}{\sigma_z^2}\right) - 1\right]^{-1}\right)$,

2. Under $H_1$, $\sqrt{T}\hat{q} \sim \mathcal{N}\left(q_0\sqrt{T}, \left[\exp\left(\frac{1}{\sigma_z^2}\right) - 1\right]^{-1}\right)$.

After normalization, we have:

1. Under $H_0$, $\sqrt{T}\hat{q}\left[\exp\left(\frac{1}{\sigma_z^2}\right) - 1\right]^{-1/2} \sim \mathcal{N}(0, 1)$,

2. Under $H_1$, $\sqrt{T}\hat{q}\left[\exp\left(\frac{1}{\sigma_z^2}\right) - 1\right]^{-1/2} \sim \mathcal{N}\left(q_0\sqrt{T}\left[\exp\left(\frac{1}{\sigma_z^2}\right) - 1\right]^{-1/2}, 1\right)$.

$\square$

# G. Proof of Theorem 3.2

Since we consider the linear regression model without intercept, which means $\boldsymbol{X}$ and $\boldsymbol{y}$ are normalized (the sum of each column is zero), we know $\frac{1}{n}\boldsymbol{J}\boldsymbol{X} = 0$ and $\frac{1}{n}\boldsymbol{J}\boldsymbol{y} = 0$, where $\boldsymbol{J} \in \mathbb{R}^{T \times n}$ is a matrix with all elements are ones. Therefore, we could rewrite the dataset generating process as

$$\tilde{\boldsymbol{X}} = (\boldsymbol{M} - \frac{1}{n}\boldsymbol{J})\boldsymbol{X} + \boldsymbol{E}_X, \quad \tilde{\boldsymbol{y}} = (\boldsymbol{M} - \frac{1}{n}\boldsymbol{J})\boldsymbol{y} + \boldsymbol{E}_Y,$$

This modification makes each element of $(\boldsymbol{M} - \frac{1}{n}\boldsymbol{J})$ a mean zero random variable for applying random matrix theory [54]. Abusing the notation, we denote $(\boldsymbol{M} - \frac{1}{n}\boldsymbol{J})$ as $\boldsymbol{M}$ in this section.

*Proof.* By definition, there holds

$$\begin{aligned}
\tilde{\boldsymbol{\beta}} &= \left[(\boldsymbol{M}\boldsymbol{X} + \boldsymbol{E}_X)^T(\boldsymbol{M}\boldsymbol{X} + \boldsymbol{E}_X)\right]^{-1}(\boldsymbol{M}\boldsymbol{X} + \boldsymbol{E}_X)^T(\boldsymbol{M}\boldsymbol{y} + \boldsymbol{E}_Y), \\
&= \left[(\boldsymbol{M}\boldsymbol{X} + \boldsymbol{E}_X)^T(\boldsymbol{M}\boldsymbol{X} + \boldsymbol{E}_X)\right]^{-1}(\boldsymbol{M}\boldsymbol{X} + \boldsymbol{E}_X)^T[\boldsymbol{M}(\boldsymbol{X}\boldsymbol{\beta}^* + \boldsymbol{\epsilon}) + \boldsymbol{E}_Y], \\
&= \left[(\boldsymbol{M}\boldsymbol{X} + \boldsymbol{E}_X)^T(\boldsymbol{M}\boldsymbol{X} + \boldsymbol{E}_X)\right]^{-1}(\boldsymbol{M}\boldsymbol{X} + \boldsymbol{E}_X)^T\left[(\boldsymbol{M}\boldsymbol{X} + \boldsymbol{E}_X)\boldsymbol{\beta}^* - \boldsymbol{E}_X\boldsymbol{\beta}^* + \boldsymbol{M}\boldsymbol{\epsilon} + \boldsymbol{E}_Y\right], \\
&= \boldsymbol{\beta}^* + \left[(\boldsymbol{M}\boldsymbol{X} + \boldsymbol{E}_X)^T(\boldsymbol{M}\boldsymbol{X} + \boldsymbol{E}_X)\right]^{-1}(\boldsymbol{M}\boldsymbol{X} + \boldsymbol{E}_X)^T(-\boldsymbol{E}_X\boldsymbol{\beta}^* + \boldsymbol{M}\boldsymbol{\epsilon} + \boldsymbol{E}_Y).
\end{aligned}$$

Therefore, there holds

$$\begin{aligned}
\|\tilde{\boldsymbol{\beta}} - \boldsymbol{\beta}^*\|_2 &= \left\|\left[(\boldsymbol{M}\boldsymbol{X} + \boldsymbol{E}_X)^T(\boldsymbol{M}\boldsymbol{X} + \boldsymbol{E}_X)\right]^{-1}(\boldsymbol{M}\boldsymbol{X} + \boldsymbol{E}_X)^T(-\boldsymbol{E}_X\boldsymbol{\beta}^* + \boldsymbol{M}\boldsymbol{\epsilon} + \boldsymbol{E}_Y)\right\|_2, \\
&\leq \left\|\left[(\boldsymbol{M}\boldsymbol{X} + \boldsymbol{E}_X)^T(\boldsymbol{M}\boldsymbol{X} + \boldsymbol{E}_X)\right]^{-1}(\boldsymbol{M}\boldsymbol{X} + \boldsymbol{E}_X)^T\right\|_2 \| - \boldsymbol{E}_X\boldsymbol{\beta}^* + \boldsymbol{E}_Y\|_2 + \cdots \\
&\quad \cdots + \left\|\left[(\boldsymbol{M}\boldsymbol{X} + \boldsymbol{E}_X)^T(\boldsymbol{M}\boldsymbol{X} + \boldsymbol{E}_X)\right]^{-1}(\boldsymbol{M}\boldsymbol{X} + \boldsymbol{E}_X)^T\boldsymbol{M}\boldsymbol{\epsilon}\right\|_2, \\
&= \sqrt{\left\|\left[(\boldsymbol{M}\boldsymbol{X} + \boldsymbol{E}_X)^T(\boldsymbol{M}\boldsymbol{X} + \boldsymbol{E}_X)\right]^{-1}\right\|_2} \| - \boldsymbol{E}_X\boldsymbol{\beta}^* + \boldsymbol{E}_Y\|_2 + \cdots \\
&\quad \cdots + \left\|\left[(\boldsymbol{M}\boldsymbol{X} + \boldsymbol{E}_X)^T(\boldsymbol{M}\boldsymbol{X} + \boldsymbol{E}_X)\right]^{-1}(\boldsymbol{M}\boldsymbol{X} + \boldsymbol{E}_X)^T\boldsymbol{M}\boldsymbol{\epsilon}\right\|_2, \quad (15)
\end{aligned}$$

where the last step is by the fact that $(\boldsymbol{A}^T\boldsymbol{A})^{-1}\boldsymbol{A}^T[(\boldsymbol{A}^T\boldsymbol{A})^{-1}\boldsymbol{A}^T]^T = (\boldsymbol{A}^T\boldsymbol{A})^{-1}$ for all suitable matrix $\boldsymbol{A}$.

We first estimate the minimal eigenvalue of $(\boldsymbol{M}\boldsymbol{X} + \boldsymbol{E}_X)^T(\boldsymbol{M}\boldsymbol{X} + \boldsymbol{E}_X)$. First of all, for the case that $n, T \to \infty$, and $0 < \lim \frac{n}{T} = \alpha < 1$, by Bai and Yin [54] on extreme eigenvalues of random matrices, there holds

$$T\frac{n-m}{mn^2}(1 - \sqrt{\alpha})^2\boldsymbol{I}_n \preceq \boldsymbol{M}^T\boldsymbol{M} \preceq T\frac{n-m}{mn^2}(1 + \sqrt{\alpha})^2\boldsymbol{I}_n, \quad (16)$$

with probability going to one when $n \to \infty$.

Secondly, in the case that $n, T \to \infty$, but $\lim \frac{n}{T} = \alpha = 0$, we will consider an expansion of $\boldsymbol{M}$, marked as $\tilde{\boldsymbol{M}} \in \mathbb{R}^{T \times (n+\tilde{n})}$, where $\tilde{n} \in \mathbb{N}$ satisfy $\lim \frac{n+\tilde{n}}{T} = 0.01$. Applying Bai and Yin [54] again, and there holds for $\tilde{\boldsymbol{M}}$ that

$$0.81T\frac{n-m}{mn^2}\boldsymbol{I}_n \preceq \tilde{\boldsymbol{M}}^T\tilde{\boldsymbol{M}} \preceq 1.21T\frac{n-m}{mn^2}\boldsymbol{I}_n, \tag{17}$$

with probability one when $n \to \infty$. Note that by definition, $\boldsymbol{M}^T\boldsymbol{M}$ is a main diagonal sub-matrix of $\tilde{\boldsymbol{M}}^T\tilde{\boldsymbol{M}}$. Thus the maximum and minimum eigenvalue of $\boldsymbol{M}^T\boldsymbol{M}$ is bounded above and below by respective eigenvalues of $\tilde{\boldsymbol{M}}^T\tilde{\boldsymbol{M}}$. Therefore, there further holds

$$0.81T\frac{n-m}{mn^2}\boldsymbol{I}_n \preceq \boldsymbol{M}^T\boldsymbol{M} \preceq 1.21T\frac{n-m}{mn^2}\boldsymbol{I}_n, \tag{18}$$

with probability one when $n \to \infty$, for the case that $\lim \frac{n}{T} = \alpha = 0$.

Therefore, combining the two cases above in Eq. equation 16 and equation 18, for all $0 \le \alpha < 1$, there holds

$$0.81T\frac{n-m}{mn}(1-\sqrt{\alpha})^2\lambda_{\min}\boldsymbol{I}_p \preceq \boldsymbol{X}^T\boldsymbol{M}^T\boldsymbol{M}\boldsymbol{X} \preceq 1.21T\frac{n-m}{mn}(1+\sqrt{\alpha})^2\lambda_{\max}\boldsymbol{I}_p, \tag{19}$$

with probability going to one when $n \to \infty$, where as a reminder, $\lambda_{\min}$ and $\lambda_{\max}$ are defined respectively to be the lower and upper bound for eigenvalues of $\boldsymbol{X}^T\boldsymbol{X}$. Apply a similar matrix expansion argument as shown above on $\boldsymbol{E}_X \in \mathbb{R}^{T \times p}$, where $\frac{p}{T} \to 0$, there holds

$$0.81\left(\frac{C_X}{m}\sigma_x\right)^2\boldsymbol{I}_p \preceq \boldsymbol{E}_X^T\boldsymbol{E}_X \preceq 1.21T\left(\frac{C_X}{m}\sigma_x\right)^2\boldsymbol{I}_p, \tag{20}$$

with probability one when $n \to \infty$. This further indicates that, with probability going to one when $n \to \infty$, there holds

$$\|\boldsymbol{E}_X^T\boldsymbol{M}\boldsymbol{X}\|_2 \le \|\boldsymbol{E}_X\|_2\|\boldsymbol{M}\boldsymbol{X}\|_2 \le 1.1(1+\sqrt{\alpha})T\sqrt{\frac{n-m}{mn}}\frac{C_X}{m}\sigma_x.$$

Note that for $m \to \infty$, there holds the following comparison on $\|\boldsymbol{E}_X^T\boldsymbol{M}\boldsymbol{X}\|_2$ and the minimal eigenvalue of $\boldsymbol{X}^T\boldsymbol{M}^T\boldsymbol{M}\boldsymbol{X}$,

$$\frac{\|\boldsymbol{E}_X^T\boldsymbol{M}\boldsymbol{X}\|_2^2}{\frac{0.81(1-\sqrt{\alpha})^2T\lambda_{\min}(n-m)}{mn}} \cdot \frac{0.81\lambda_{\min}(1-\sqrt{\alpha})^2}{1.21(1+\sqrt{\alpha})^2} \le \frac{mn\left(\frac{C_X}{m}\sigma_x\right)^2}{n-m} = \frac{nC_X^2}{m(n-m)\ln\left(1+\frac{\mu^2}{\nu^2}\right)} \to 0,$$

where the second last step is because: $\frac{m^2T}{n^2} \to \nu$ and $T \to \infty$ leads to $\frac{n}{n-m} \to 1$; $m \to \infty$ leads to $\frac{1}{m} \to 0$. Therefore, with probability going to one when $n \to \infty$, there holds

$$(\boldsymbol{M}\boldsymbol{X} + \boldsymbol{E}_X)^T(\boldsymbol{M}\boldsymbol{X} + \boldsymbol{E}_X) \succeq 0.81\frac{T\lambda_{\min}(n-m)}{mn}(1-\sqrt{\alpha})^2\boldsymbol{I}_p - 2\|\boldsymbol{E}_X^T\boldsymbol{M}\boldsymbol{X}\|_2 \cdot \boldsymbol{I}_p, \tag{21}$$

$$\succeq \frac{T\lambda_{\min}(n-m)}{2mn}(1-\sqrt{\alpha})^2\boldsymbol{I}_p. \tag{22}$$

Therefore, there holds

$$\sqrt{\left\|[(\boldsymbol{M}\boldsymbol{X}+\boldsymbol{E})^T(\boldsymbol{M}\boldsymbol{X}+\boldsymbol{E})]^{-1}\right\|_2} \le \left[(1-\sqrt{\alpha})\sqrt{\frac{T\lambda_{\min}(n-m)}{2mn}}\right]^{-1}, \tag{23}$$

with probability going to one when $n \to \infty$.

What's more, using Eq. equation 20 and central limit theorem, there holds

$$\|-\boldsymbol{E}_X\beta^* + \boldsymbol{E}_Y\|_2 \le 1.1\sqrt{T}\frac{C_X}{m}\sigma_x\|\boldsymbol{\beta}^*\|_2 + 1.1\sqrt{T}\frac{C_Y}{m}\sigma_y,$$

with probability going to one when $n \to \infty$. Therefore, there holds

$$\sqrt{\left\|\left[(\boldsymbol{MX} + \boldsymbol{E}_X)^T(\boldsymbol{MX} + \boldsymbol{E}_X)\right]^{-1}\right\|_2} \| - \boldsymbol{E}_X \boldsymbol{\beta}^* + \boldsymbol{E}_Y \|_2 \tag{24}$$

$$\leq \frac{2C_X \|\boldsymbol{\beta}^*\|_2 + 2C_Y}{\sqrt{\lambda_{\min}}(1 - \sqrt{\alpha})} \frac{1}{\sqrt{\frac{m(n-m)}{n} \ln\left(1 + \frac{\mu^2 n^2}{m^2 T}\right)}}, \tag{25}$$

with probability going to one when $n \to \infty$. Note that $m^2 T/n^2 \to \nu^2$ and $T \to \infty$ imply that $(n-m)/n = 1 - \nu/\sqrt{T} \to 1$ and $m = \nu n/\sqrt{T}$, therefore

$$bias \leq \frac{T^{1/4}}{n^{1/2}} \frac{2C_X \|\boldsymbol{\beta}^*\|_2 + 2C_Y}{\sqrt{\lambda_{\min}}(1 - \sqrt{\alpha})} \cdot \frac{1}{\sqrt{\nu \ln\left(1 + \frac{\mu^2}{\nu^2}\right)}}. \tag{26}$$

Moreover, consider taking expectation on $\boldsymbol{\epsilon}$ only, there holds

$$\mathbb{E}_{\boldsymbol{\epsilon}}\left(\left[(\boldsymbol{MX} + \boldsymbol{E}_X)^T(\boldsymbol{MX} + \boldsymbol{E}_X)\right]^{-1}(\boldsymbol{MX} + \boldsymbol{E}_X)^T \boldsymbol{M}\boldsymbol{\epsilon}\right) = 0,$$

$$\text{var}_{\boldsymbol{\epsilon}}\left(\left[(\boldsymbol{MX} + \boldsymbol{E}_X)^T(\boldsymbol{MX} + \boldsymbol{E}_X)\right]^{-1}(\boldsymbol{MX} + \boldsymbol{E}_X)^T \boldsymbol{M}\boldsymbol{\epsilon}\right)$$

$$= \sigma^2 \left[(\boldsymbol{MX} + \boldsymbol{E}_X)^T(\boldsymbol{MX} + \boldsymbol{E}_X)\right]^{-1}(\boldsymbol{MX} + \boldsymbol{E}_X)^T \boldsymbol{M}\boldsymbol{M}^T(\boldsymbol{MX} + \boldsymbol{E}_X)\left[(\boldsymbol{MX} + \boldsymbol{E}_X)^T(\boldsymbol{MX} + \boldsymbol{E}_X)\right]^{-1}$$

$$\preceq 1.21 T \frac{n-m}{mn^2}\left[(\boldsymbol{MX} + \boldsymbol{E}_X)^T(\boldsymbol{MX} + \boldsymbol{E}_X)\right]^{-1}(\boldsymbol{MX} + \boldsymbol{E}_X)^T I_n (\boldsymbol{MX} + \boldsymbol{E}_X)\left[(\boldsymbol{MX} + \boldsymbol{E}_X)^T(\boldsymbol{MX} + \boldsymbol{E}_X)\right]^{-1}$$

$$= 1.21 T \frac{n-m}{mn^2}\left[(\boldsymbol{MX} + \boldsymbol{E}_X)^T(\boldsymbol{MX} + \boldsymbol{E}_X)\right]^{-1}$$

$$\preceq \frac{4\sigma^2}{n} \frac{(1 + \sqrt{\alpha})^2 \lambda_{\max}}{(1 - \sqrt{\alpha})^2 \lambda_{\min}} I_p,$$

with probability going to one when $n \to \infty$, where the last step is from Eq. equation 23.

For simplicity of notations, denote $\zeta = \left[(\boldsymbol{MX} + \boldsymbol{E}_X)^T(\boldsymbol{MX} + \boldsymbol{E}_X)\right]^{-1}(\boldsymbol{MX} + \boldsymbol{E}_X)^T \boldsymbol{M}\boldsymbol{\epsilon}$. Note that $\zeta_i$ is Gaussian random variable for any given $\boldsymbol{M}$ and $\boldsymbol{E}_X$, therefore, with probability going to one when $n \to \infty$, there holds

$$\lim_{n \to \infty} \mathbb{P}_{\boldsymbol{\epsilon}}\left(|\zeta_i| < \frac{2\sigma \ln n}{\sqrt{n}} \frac{(1 + \sqrt{\alpha})\sqrt{\lambda_{\max}}}{(1 - \sqrt{\alpha})\sqrt{\lambda_{\min}}}\right) = 1.$$

Therefore, there further holds

$$\mathbb{P}_{\boldsymbol{\epsilon}}\left(\left\|\left[(\boldsymbol{MX} + \boldsymbol{E}_X)^T(\boldsymbol{MX} + \boldsymbol{E}_X)\right]^{-1}(\boldsymbol{MX} + \boldsymbol{E}_X)^T \boldsymbol{M}\boldsymbol{\epsilon}\right\|_2 > \frac{2\sigma\sqrt{p} \ln n}{\sqrt{n}} \frac{(1 + \sqrt{\alpha})\sqrt{\lambda_{\max}}}{(1 - \sqrt{\alpha})\sqrt{\lambda_{\min}}}\right)$$

$$\leq \sum_{i=1}^{p} \mathbb{P}_{\boldsymbol{\epsilon}}\left(|\zeta_i| > \frac{2\sigma \ln n}{\sqrt{n}} \frac{(1 + \sqrt{\alpha})\sqrt{\lambda_{\max}}}{(1 - \sqrt{\alpha})\sqrt{\lambda_{\min}}}\right) \to 0,$$

with probability going to one when $n \to \infty$. Therefore, there holds

$$\left\|\left[(\boldsymbol{MX} + \boldsymbol{E}_X)^T(\boldsymbol{MX} + \boldsymbol{E}_X)\right]^{-1}(\boldsymbol{MX} + \boldsymbol{E}_X)^T \boldsymbol{M}\boldsymbol{\epsilon}\right\|_2 \leq \frac{2\sigma\sqrt{p} \ln n}{\sqrt{n}} \frac{(1 + \sqrt{\alpha})\sqrt{\lambda_{\max}}}{(1 - \sqrt{\alpha})\sqrt{\lambda_{\min}}}, \tag{27}$$

with probability going to one when $n \to \infty$.

Combining Eq. equation 15, Eq. equation 27 and Eq. equation 26 together, and we will have our final result. $\qquad\square$

## H. Inconsistency of $\tilde{\beta}$ with bounded $m$

We show in this section that $m \to \infty$ is necessary for the estimator $\tilde{\beta}$ defined by

$$\tilde{\beta} = \left[\tilde{X}^T \tilde{X}\right]^{-1} \tilde{X}^T \tilde{y},$$

to be consistent.

The bias term of the estimator $\tilde{\beta}$ is

$$\mathbb{E}_{\epsilon}[\tilde{\beta}] - \beta^* = \left[(MX + E_X)^T(MX + E_X)\right]^{-1}(MX + E_X)^T(-E_X\beta^* + E_Y).$$

Specifically, we consider the following term in the bias term, that we consider

$$\left[(MX + E_X)^T(MX + E_X)\right]^{-1}E_X^T E_X\beta^*, \tag{28}$$

and we will show that such a term will not converge to zero if $m$ is bounded.

Note that by Eq. equation 19 and Eq. equation 20 that

$$(MX + E_X)^T(MX + E_X) \preceq 2X^T M^T M X + 2E_X^T E_X,$$

$$\preceq 3T\left(\frac{n-m}{mn}(1 + \sqrt{\alpha})^2\lambda_{\max} + \frac{C_X^2}{m^2\ln(1 + \mu^2/\nu^2)}\right)I_p,$$

$$\preceq 4T\left(\frac{1}{m}(1 + \sqrt{\alpha})^2\lambda_{\max} + \frac{C_X^2}{m^2\ln(1 + \mu^2/\nu^2)}\right)I_p,$$

with probability goes to one when $n \to \infty$. Then, combining with Eq. equation 20 again, there holds with probability goes to one when $n \to \infty$,

$$\|\left[(MX + E_X)^T(MX + E_X)\right]^{-1}E_X^T E_X\beta^*\|_2$$

$$\geq \frac{1}{8}\left(\frac{1}{m}(1 + \sqrt{\alpha})^2\lambda_{\max} + \frac{C_X^2}{m^2\ln(1 + \mu^2/\nu^2)}\right)^{-1}\frac{C_X^2}{m^2\ln(1 + \mu^2/\nu^2)}\|\beta^*\|_2,$$

which clearly will not goes to zero if $m$ is bounded, and the estimator $\tilde{\beta}$ will be inconsistent.

# I. Details of Experiments

## I.1. Datasets

MNIST [19] has 10 categories and each category has 6000 and 1000 28×28 grayscale images for training and testing. The colored images of CIFAR10 and CIFAR100 [20] have a spatial size of 32×32. CIFAR10 owns 10 categories, each with 5000 and 1000 RGB images for training and testing, respectively. CIFAR100 owns 100 categories, each with 500 and 100 RGB images for training and testing, respectively. MiniImagenet [21] dataset contains 60000 samples and 100 classes. For each class, set 500 samples for training and 100 samples for evaluation. All datasets are publicly available and widely used in the machine learning community.

## I.2. Detailed Settings for our methods and Baselines

**DPMix**: We tried different networks for DPMix. When comparing the utility of different privacy accountants in Table 6, we use the same network described by Lee et al. [6] to show that GDP accountant could improve the results of Lee et al. [6]. When comparing NeuroMixGDP, we choose the same ScatteringNet as NeuroMixGDP for MNIST to make a fair comparison. We chose the same CNN described in Lee et al. [6] for CIFAR10/100, because we tried ResNet-50 and ResNet-152 for DPMix on CIFAR10/100 dataset in Tables 1 and 7 and showed larger networks lead to the weaker utility because of overfitting. The experiments in Tables 1 and 7 use modified ResNet-50 and ResNet-152, which have an input size of 32*32*3 to keep the dimension for pixel mixup the same as the network described in Lee et al. [6] to make a fair comparison. While the modification of input size also improves the utility of pre-trained ResNets with 224*224*3 input size. For example, the best test CIFAR10 accuracy during training is only 15.64% for pre-trained ResNet-50 with $(8, 10^{-5})$-DP. If not specified, we set $C_x = C_y = 1$. We train the model with Adam for 200 epochs and batch size 256. The initial learning rate is 0.1 and decays by 10 at epochs 80,120,160.

**NeuroMixGDP andNeuroMixGDP-HS**: When we conduct mixup at the feature level, we build a linear classifier upon the mixed features. For MNIST, which contains gray-scale images, we use ScatteringNet [18] described in Appendix I.3 as the feature extractor and the output size of Scattering Net is 81*7*7. For RGB images, we choose ResNet-50 and ResNet-152 pre-trained on unlabeled ImageNet with SimCLRv2 as a feature extractor [55]. To fit the input size, we resize the images to 224*224 and use features with 2048 and 6144 dimensions before the final classification layer. We set $C_x = C_y = 1$ in all experiments. We train the model with Adam for 200 epochs and batch size 256. The initial learning rate is 0.001 and decays by 10 at epoch 80,120 and, 160.

**Non-private baseline**: Training without considering privacy. The setting of training is the same as the corresponding methods except for no mixup or noise. It serves as an upper bound of the utility of DP methods.

**P3GM**: P3GM [5] is a data publishing algorithm that first trains a VAE with DPSGD, then releases samples generated from the DP VAE. We use the officially released code of P3GM and follow its default setting. The code only offers the hyper-parameter setting for $(1, 10^{-5})$-DP. To evaluate P3GM with different privacy budgets, we only alter the noise scale and keep other hyper-parameter fixed. After getting the generated samples, we use the same ScatteringNet classifier as NeuroMixGDP for a fair comparison on MNIST dataset. For CIFAR10/100 datasets, we choose the CNN described by Lee et al. [6]. We choose the same optimization configuration except for setting the learning rate to 0.0001, which offers the best utility for P3GM.

**DP-MERF** DP-MERF [26] first calculates random feature representations of kernel mean embeddings with DP noise and then trains a generator to produce fake samples that match the noisy embeddings. We use the code from `https://github.com/ParkLabML/DP-MERF` and its default hyper-parameter setting except change the noise scale to achieve different privacy guarantees.

**DP-HP** DP-HP [28] improves DP-MERF by replacing the random features with Hermite polynomial features, which can use fewer dimensions to accurately approximate the mean embedding of the data distribution compared to random features. Therefore, DP-HP injects less DP noise and is more suitable for releasing private data. We use the code from `https://github.com/ParkLabML/DP-HP` and its default hyper-parameter setting except change the noise scale to achieve different privacy guarantees.

**Generative model for feature release.** We also provide classification results of P3GM, DP-MERF, and DP-HP on the features extracted by ScatteringNet or ResNet-152 pre-trained on unlabeled ImageNet with SimCLRv2. Specifically, we first conduct feature extraction and then train the generative model in the features space to generate a synthetic feature dataset. Finally, we build a linear classifier on the synthetic feature dataset with the same setting as the one used for NeuroMixGDP(-HS).

## I.3. Network Structure

In our experiments, we employed the same CNN as Lee et al. [6]. It contains 357,690 parameters, and here we list its detailed structure.

Table 5: CNN model used for MNIST, CIFAR10, and CIFAR100, with ReLU activation function.

| Layer type | Configuration |
|---|---|
| Conv2d | 32 filters of 5x5, stride 1, padding 2 |
| Max-Pooling | 2x2, stride 2 |
| Conv2d | 48 filters of 5x5, stride 1, padding 2 |
| Max-Pooling | 2x2, stride 2 |
| Fully connected | 100 units |
| Fully connected | 100 units |
| Fully connected | 10 units |

As for the ScatteringNet [56], we follow the setting of Tramer and Boneh [18] for all experiments. We set Scattering Network of depth $J = 2$. And use wavelet filters rotated along eight angles. Therefore,

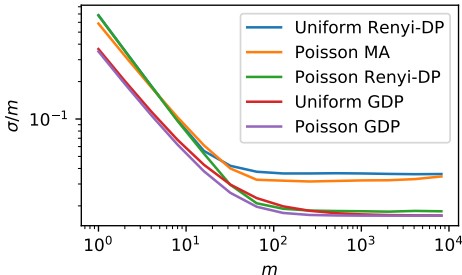

Figure 10: Comparison of noise injected to achieve $(1, 10^{-5})$-DP by different privacy accountants.

for input images with size $H \times W$, the feature maps after transformation are $(81, H/4, W/4)$. We also conducted group normalization [57] with 27 groups to the features extracted by ScatteringNet. Then a linear classifier will be built on the ScatteringNet feature. We use Kymotio `https://www.kymat.io/` under the 3-Clause BSD License for implementation.

Since there are too many variants, we also provide detailed configurations of ResNet-50 and ResNet-152. ResNet-50 in our paper has 24M parameters and corresponds to the "ResNet-50 1X without Selective Kernel" in Chen et al. [55]. ResNet-152 has 795M parameters and corresponds to the "ResNet-152 3X with Selective Kernel" in Chen et al. [55]. The checkpoint of the corresponding model can be downloaded from `gs://simclr-checkpoints/simclrv2/pretrained` under Apache License 2.0.

## I.4. GDP with Poisson Sampling Improves Utility.

We give a comparison of different subsampling schemes and accountants. Under typical setting of NeuroMixGDP where $T = n = 50000$, $\epsilon = 1$, $\delta = 10^{-5}$, and $\sigma_x = \sigma_y$, we show the noise $\sigma_x/m$ vs. $m$, injected by "Uniform RDP", which is used by Lee et al. [6], "Poisson MA" [58], "Poisson RDP" [49], "Uniform GDP" [17], and "Poisson GDP" (ours) in Fig. 10. We can see that our analysis uses the smallest noise for all $m$ and is thus the tightest.

Table 6: Test accuracy (%) of DPMix and DPMix with GDP (DPMix-GDP).

|  | MNIST | | CIFAR10 | |
| --- | --- | --- | --- | --- |
|  | DPMix | DPMix-GDP | DPMix | DPMix-GDP |
| $\epsilon = 1$ | 65.82±5.48 | 79.19±0.38 | 18.64±1.79 | 24.36±1.25 |
| $\epsilon = 2$ | 77.40±2.28 | 82.71±1.33 | 24.12±3.42 | 30.37±1.30 |
| $\epsilon = 4$ | 81.13±1.51 | 84.09±0.70 | 29.88±1.04 | 31.97±0.71 |
| $\epsilon = 8$ | 84.41±0.90 | 85.71±0.28 | 31.27±0.90 | 32.92±1.03 |

We also compare DPMix with Uniform sub-sampled Renyi-DP (DPMix) and Poisson subsampled GDP (DPMix-GDP) in Table 6. The detailed settings are listed in Appendix I.2. We report the best accuracy with optimal $m^*$. We can see that in all cases, DPMix-GDP is better than DPMix with RDP. In particular, the advantage of DPMix-GDP becomes larger when the privacy budget is tighter. For a fair comparison, we choose GDP as the privacy accountant and report GDP privacy guarantee for DPMix, DPSGD, P3GM, and NeuroMixGDP(-HS).

## I.5. Experiments about Neural Collapse and Minimum Euclidean Distance

First, we give a formal definition of NC metric. Let $x_i$ be the feature, $\mu_G$ be the global mean of features and $\mu_k$ is the mean of features of class $k$. The within-class covariance is defined as:

$$\Sigma_W := \frac{1}{n} \sum_{i \in [n]} (x_i - \mu_{y_i})(x_i - \mu_{y_i})^T \tag{29}$$

The between-class covariance is defined as:

$$\Sigma_B := \frac{1}{K} \sum_{k \in [K]} (\mu_k - \mu_G)(\mu_k - \mu_G)^T \tag{30}$$

The NC metric used in Papyan et al. [9] is $Tr(\Sigma_W \Sigma_B^\dagger / K)$ where $Tr$ is the trace operator, $K$ is the total number of classes, and $\dagger$ is Moore–Penrose pseudoinverse.

$Tr(\Sigma_W \Sigma_B^\dagger / K)$ is an indicator of the collapse of within-class variation. When comparing $Tr(\Sigma_W \Sigma_B^\dagger / K)$ we clipped the input images and feature of CIFAR10/100 to make them have the same $l_2$-norm. ScatteringNet feature extractor is designed for a variety of invariant properties, e.g. (local) translation/rotation/deformation invariant, which will also reduce $Tr(\Sigma_W \Sigma_B^\dagger / K)$. We observe that the ScatteringNet reduces the NC metric from 1.53 to 1.16 for MNIST dataset.

We also calculate the MED for the CIFAR datasets following the experiment setting as described in Sec 2.4 of [8]. We first clip both features and raw images with $C_x = 1$. Then we conduct both input and feature mixup with $m = 2$ and $T = n = 50000$. Finally, we compute the minimum distance between each mixed sample with clean samples with different labels. This MED is an estimation of "$\epsilon$" defined in Figure 6 of [8], which is used to characterize the degree of AC. Larger MED corresponds to weaker AC.

### I.6. Source of Empirical Results in Table 2

Results of DP-GM, PrivBayes and Ryan's are from Table VII in Takagi et al. [5]. Results of DP-CGAN, DP-MERF and DP-Sinkhorn are from Table 1 in Cao et al. [25]. Results of DP-GAN, Pate-GAN, G-PATE, GS-WGAN and DataLens are from Table 1 in Wang et al. [24].

## J. More Empirical Experiments

### J.1. Validating Theorem 3.2 through simulation

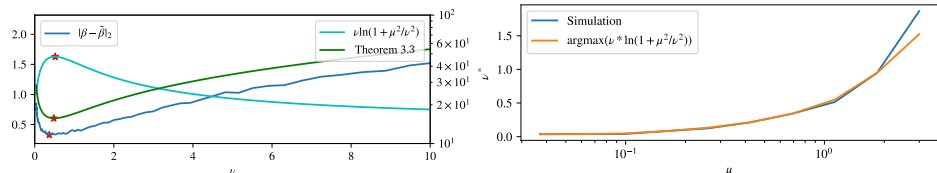

Figure 11: Comparison of optimal $\nu^*$ from simulation and Theorem 3.2. Note that the optimal $\nu^*$ corresponds to the optimal mixup degree $m^*$ for fixed $n$ and $T$.

We use simulation experiments to validate Theorem 3.2 is tight for characterizing optimal $\nu^*$. Let $X \in \mathbb{R}^{n \times 5}$ and each row of $X$ is sampled from $\sim \mathcal{N}(0, \Sigma_X)$, where $n = 300$ and the covariance matrix $\Sigma_{X i,j} = 0.3^{|i-j|}$. We sample $\beta \in \mathbb{R}^{5 \times 1}$ from $\sim \mathcal{N}(0, I)$. Let $Y = X\beta + \epsilon$ and each element $\epsilon_i \sim \mathcal{N}(0, 1)$. Each column of $X$ and $Y$ is subtracted by its mean to ensure that each column's sum is zero. For NeuroMixGDP, we set $T = 3n$, and $\mu = 1$. We choose $C_x = \max_i \|X_i\|_2, C_y = \max_i \|Y_i\|_2$ such that the clipping operation never takes effect. We run simulation experiments with different $m$ and compare the estimation error $\|\beta - \tilde{\beta}\|_2$ with Theorem 3.2 in Fig. 11 (Upper). Fig. 11 Lower studies the effect of $\mu$ to $\nu^*$ by changing $\mu \in [3^{-3}, 3]$. We report medians of 20 independent simulations.

### J.2. Hyper-parameter Tuning of NeuroMixGDP

**Keeping a balance between noise on features and labels by tuning $\lambda$:** The detailed grid search results are shown in Appendix Fig. 12. There are different optimal $\lambda$ for each setting. MNIST and CIFAR10 prefer $\lambda \in [1, 4]$. CIFAR100 dataset prefers $\lambda \in [0.5, 2]$ since CIFAR100 has more categories and will be more sensitive to the noise on labels. When searching optimal $\lambda$ is not feasible in

practice, we think $\lambda \in [1, 2]$ could be a good choice. For example, comparing with optimal settings, the maximal loss on test accuracy is 1.49%, 1.08%, and 0.46% for MNIST, CIFAR10, and CIFAR100, respectively when we set $\lambda = 1$.

**Choosing mixup degree** $m$**:** One feature of DPMix is that it mixup $m$ samples to reduce the operation's sensitivity and the DP noise. We conduct a gird search to find optimal $m$ with $\lambda = 1$; the results are shown in Fig. 3. There are three key observations. First, in all cases, including DPMix and NeuroMixGDP, the utility has an increasing-decreasing trend, and there is an optimal $m^*$ that can maximize the utility. Second, for the same task, smaller privacy budgets tend to have smaller optimal $m$. Third, when tuning $m$ is not feasible, setting an empirical value for $m$ still works and will not lose too much utility. From Fig. 3, we can see that for all three datasets, the $m^*$ is between 32 and 256. We may choose $m = 64$ to be an empirical choice. Then the maximal loss in test accuracy is 0.74%, 0.46%, and 3.40% for MNIST, CIFAR10, and, CIFAR100, respectively, and that should be accepted.

Table 7: CIFAR10 test accuracy (%) of DPMix and NeuroMixGDP (Ours) with different models. We show the convergence test accuracy before "+" and the improvement of reporting best accuracy during training after "+".

| $\epsilon$ | 1 | 2 | 4 | 8 | Non-private |
|---|---|---|---|---|---|
| DPMix | 24.36+5.60 | 30.37+2.07 | 31.97+1.72 | 32.92+3.01 | 79.22 |
| DPMix ResNet-50 | 14.48+10.62 | 18.73+7.98 | 12.61+8.69 | 15.52+16.19 | 94.31 |
| DPMix ResNet-152 | 20.77+3.99 | 17.54+11.52 | 17.06+12.50 | 15.35+15.31 | 95.92 |
| Ours ResNet-50 | 75.58+0.36 | 77.31+0.20 | 79.02+0.14 | 80.75+0.04 | 94.31 |
| Ours ResNet-152 | 87.74+0.29 | 89.01+0.13 | 90.21+0.11 | 90.83+0.04 | 95.92 |

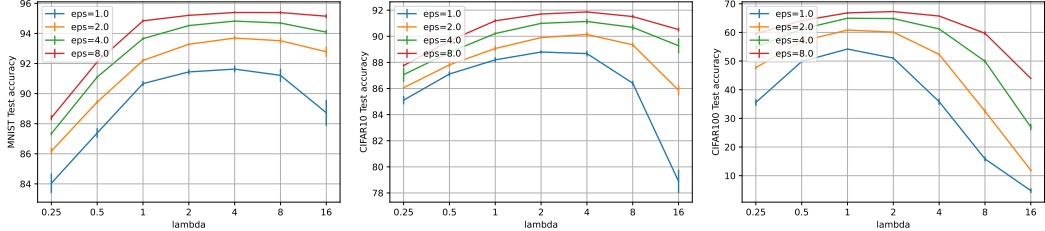

Figure 12: These figures show how utility changed w.r.t. $\lambda$ under different settings.

Table 8: The table shows MNIST test accuracy (%) of DPMix using CNN or ScatteringNet as classifier, we show the best test accuracy with $m^*$. ScatteringNet classifier performs better

| | DPMix with CNN | DPMix with ScatteringNet |
|---|---|---|
| $\epsilon = 1$ | 79.19$\pm$0.38 | 82.16$\pm$0.54 |
| $\epsilon = 2$ | 82.71$\pm$1.33 | 83.28$\pm$0.53 |
| $\epsilon = 4$ | 84.09$\pm$0.70 | 84.90$\pm$0.26 |
| $\epsilon = 8$ | 85.71$\pm$0.28 | 86.00$\pm$0.28 |

## J.3. Membership inference attack against NeuroMixGDP and NeuroMixGDP-HS

Due to space limitations, we are unable to include the complete results of the membership inference attack in the main text. However, in this section, we provide a comprehensive presentation of the attack results in Table 9, which also includes a performance comparison of DPSGD. It is important to note that all the methods listed in Table 9 offer a differential privacy (DP) guarantee, resulting in

their achieved Area Under the Curve (AUC) values being very close to perfect protection, i.e., 0.5. Consequently, smaller privacy budgets correspond to reduced utility but also smaller AUC values. Notably, in the low $\epsilon$ range (i.e., $\epsilon \leq 0.5$), NeuroMixGDP-HS demonstrates favorable utility while maintaining an AUC that is almost on par with perfect protection.

Table 9: Membership inference on CIFAR100

|  | DPSGD | | NeuroMixGDP | | NeuroMixGDP-HS | |
|---|---|---|---|---|---|---|
|  | Test Acc | AUC | Test Acc | AUC | Test Acc | AUC |
| $\epsilon = 8$ | 72.77±0.15 | 0.5097±0.0002 | 67.15±0.29 | 0.5078±0.0004 | 73.39±0.20 | 0.5117±0.0001 |
| $\epsilon = 4$ | 72.40±0.14 | 0.5096±0.0002 | 64.32±0.11 | 0.5073±0.0006 | 72.54±0.08 | 0.5117±0.0004 |
| $\epsilon = 2$ | 71.37±0.22 | 0.5092±0.0003 | 60.61±0.42 | 0.5059±0.0007 | 71.88±0.14 | 0.5107±0.0003 |
| $\epsilon = 1$ | 68.13±0.39 | 0.5091±0.0009 | 54.32±0.50 | 0.5052±0.0014 | 71.72±0.09 | 0.5090±0.0003 |
| $\epsilon = 0.5$ | 59.53±0.68 | 0.5052±0.0012 | 43.21±0.84 | 0.5048±0.0008 | 70.23±0.04 | 0.5096±0.0004 |
| $\epsilon = 0.2$ | 38.90±0.50 | 0.5039±0.0016 | 12.54±0.58 | 0.5015±0.0011 | 65.78±0.41 | 0.5066±0.0004 |
| $\epsilon = 0.1$ | 18.25±0.58 | 0.5039±0.0009 | 2.72±0.46 | 0.4984±0.0026 | 57.53±0.64 | 0.5056±0.0009 |

## J.4. Comparing DPSGD and NeuraMixGDP-HS on the ImageNet dataset

the ImageNet dataset which contains 1.28M samples of 1000 classes. We set $\delta = 1e - 7$ and report test accuracy with different $\epsilon$ in the Table below. We only evaluate CLIP feature extractor since SimCLRv2 is pretrained on the ImageNet dataset. The Table shows that our methods achieve comparable utility to DPSGD when $\epsilon$ is large and much better utility when $\epsilon$ is smaller, demonstrating our potential of releasing large-scale datasets.

Table 10: Comparing DPSGD and NeuraMixGDP-HS on the ImageNet dataset

|  | $\epsilon = 0.1$ | $\epsilon = 0.2$ | $\epsilon = 0.5$ | $\epsilon = 1.0$ | $\epsilon = 2.0$ |
|---|---|---|---|---|---|
| DPSGD | 59.07±1.25 | 72.06±0.42 | 74.51±0.59 | 76.90±0.31 | 77.40±0.21 |
| Ours | 74.21±0.34 | 76.16±0.36 | 77.30±0.24 | 77.50±0.30 | 77.97±0.18 |

## J.5. Avg-Mix out performs RW-Mix on CIFAR10 dataset

Fig. 1 shows CIFAR100 test accuracy of Avg-Mix and RW-Mix with $(\epsilon = 1, \delta = 1e - 5)$-DP. We set $C_x = C_y = 1, T = 50000$ and conduct a grid search on $m \in [1, 4096]$ and $\lambda \in [0.25, 16]$. We also conduct experiments on CIFAR10 dataset with above settings. Fig. 13 shows Avg-Mix consistently out performs RW-Mix duo to lower sensitivity on CIFAR10 dataset.

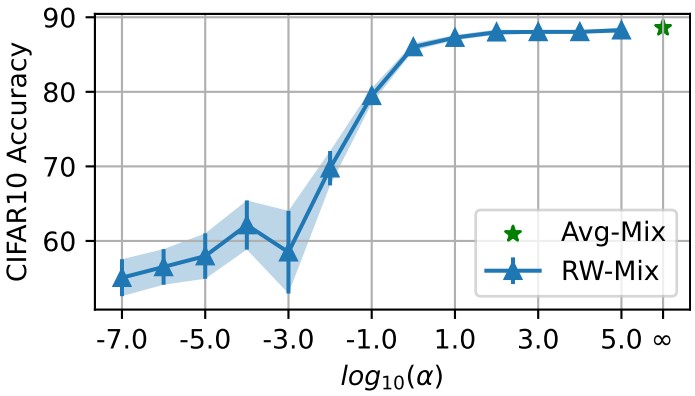

Figure 13: Avg-Mix out performs RW-Mix on CIFAR10 dataset

### J.6. More cases of Figure 2

Figure 2 shows the distribution of mixed label $\tilde{y}$. We conduct experiments by varing the mixup degree $m \in [128, 256, 512, 1024, 2048, 4096]$ and class sample rate $p \in 0.10.30.5$ on CIFAR10 dataset. We find in all cases, $\tilde{y}$ from Avg-Mix PS are concentrated on $1/K$ and overwhelmed by DP noise while HS produces strong class components, which are distinguishable from DP noise.

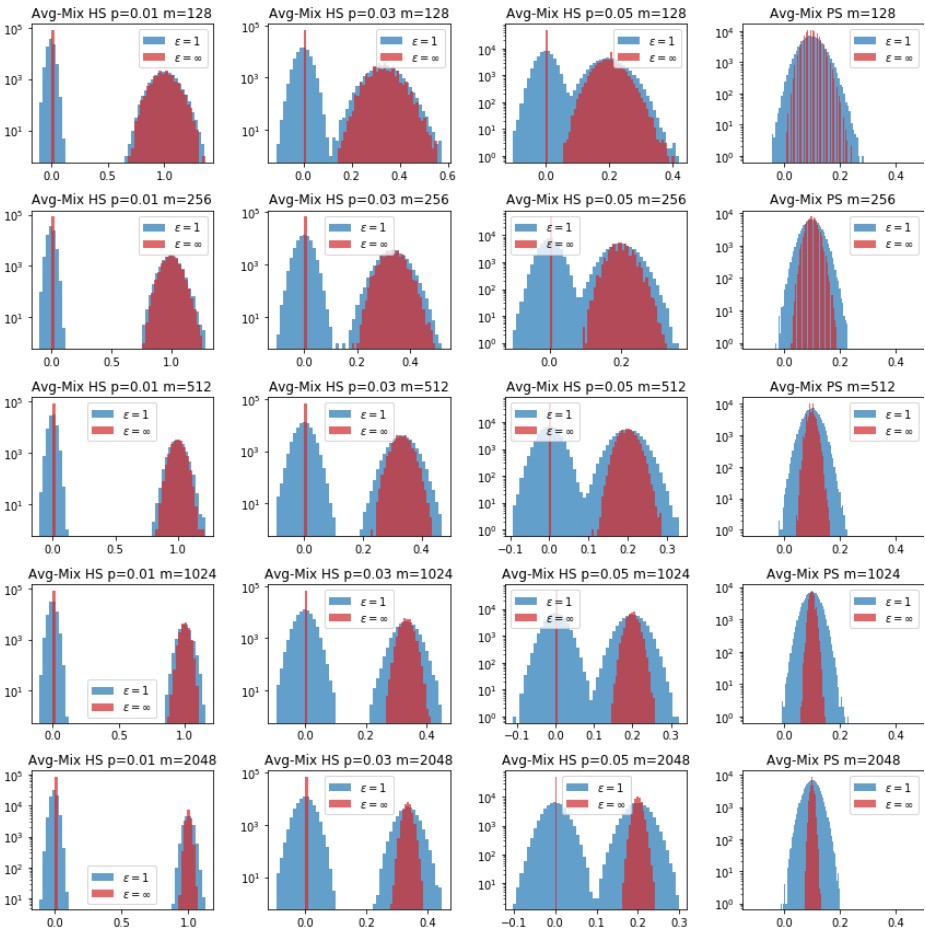

Figure 14: More cases of Figure 2.

# K. Model Inversion Attack

This section introduces the background of model inversion attacks and implementation details. Some visualization results are also shown here. He et al. [31] first introduced regularized Maximum Likelihood Estimation (rMLE) to recover the input data in the split learning framework. Under the white-box attack setting, the attackers know intermediate features and the neural network with parameters. Since NeuroMixGDP(-HS) also releases features and does not assume the feature extractor is private, white-box MIA is a proper attack method for NeuroMixGDP(-HS). We will modify the attack algorithm to adopt mixup setting and show we can defend this type of attack using extensive experiments. Let us first recall the white-box attack by He et al. [31]. Let $\boldsymbol{x}_0$ be the original data; the original MIA uses an optimization algorithm to find $\hat{x}$ as the recovered input.

$$\hat{\boldsymbol{x}} = \arg\min_{\boldsymbol{x}} \|f_1(\theta_1, \boldsymbol{x}_0) - f_1(\theta_1, \boldsymbol{x})\|_2^2 + \lambda TV(\boldsymbol{x})$$

Where $f_1(\theta_1, \cdot)$ is the feature extractor, $TV$ is the total variation loss proposed in Rudin et al. [59].

For simplicity, we denote the norm clipping and average process as $\bar{f}_1$. For example, $\bar{f}_1(\theta_1, \boldsymbol{x}_0)$ represent $\frac{1}{m}\sum_{i=1}^{m}\left(f_1(\theta_1, x_0^{(i)})/\max(1, \|f_1(\theta_1, x_0^{(i)})\|_2/C)\right)$. Let $\bar{TV}(\boldsymbol{x})$ denotes $\frac{1}{m}\sum_{i=1}^{m}TV(\boldsymbol{x}^{(i)})$. Then, recall that $\bar{\boldsymbol{x}}_t$ is in the released feature in NeuroMixGDP(-HS), then the following attack could use the following process to recover the private data.

$$\hat{\boldsymbol{x}} = \arg\min_{\boldsymbol{x}} \|\bar{\boldsymbol{x}}_t - \bar{f}_1(\theta_1, \boldsymbol{x})\|_2^2 + \lambda\bar{TV}(\boldsymbol{x})$$

We implement such an attack on MNIST and CIFAR10 datasets with ScatteringNet features since MIA towards a larger model like ResNet-152 could be hard even without feature mixup and DP noise [31] (See Appendix Fig. 15 for MIA on ResNet-152 features). We use random images sampled from uniform distribution as the initialization of $x$ and use Adam [60] with $\lambda = 0.0001$, the learning rate of 0.1, and maximal steps of 5000 to obtain $\hat{\boldsymbol{x}}$.

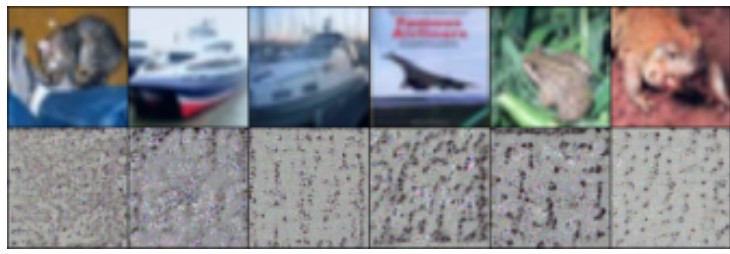

Figure 15: Attacking from features without mixup and DP noise of ResNet-152 on CIFAR10 dataset. MIA fails to attack such a deep model with 152 layers. This observation meets the one in He et al. [31]

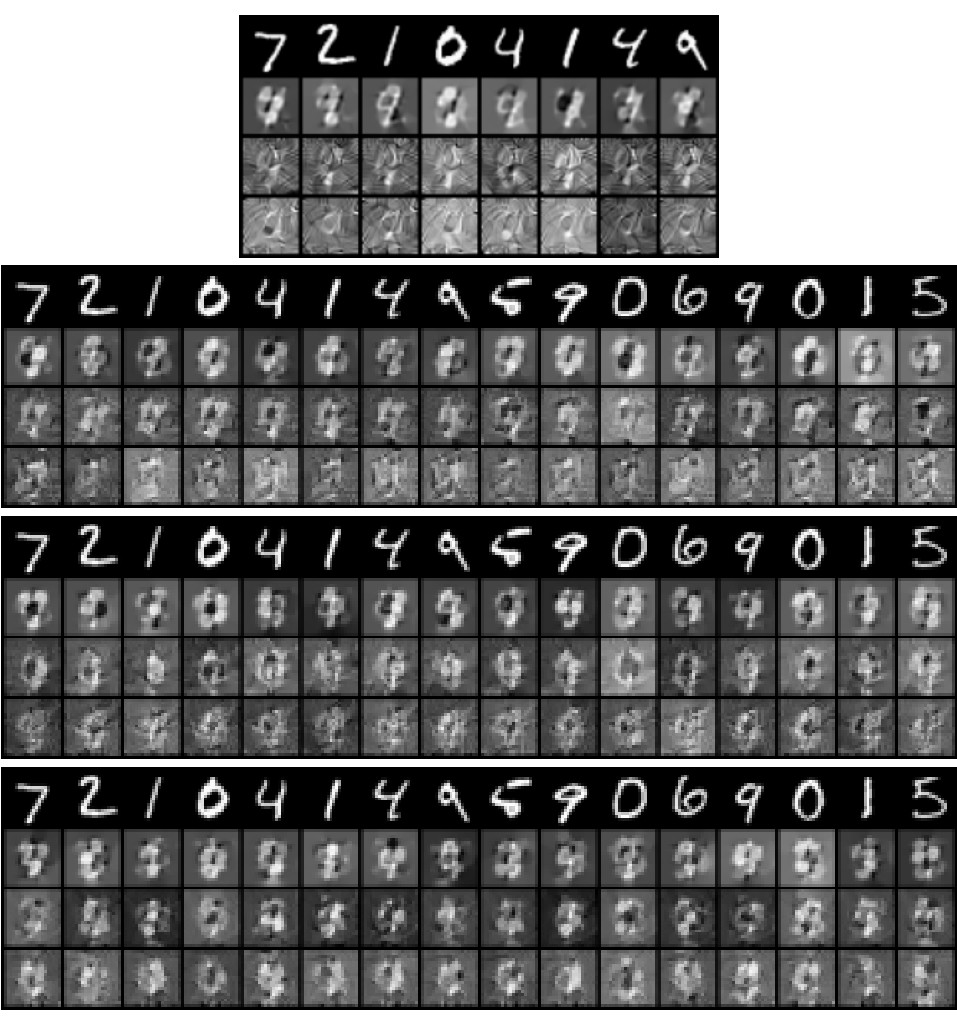

Figure 16: Attacking from features of Scattering net on MNIST dataset. The first row corresponds to raw images, and the following three correspond to recovered images by MIA with $\sigma = 0$, $\epsilon = 8$, and $\epsilon = 1$, respectively. Due to the limited space, we present $m = 8, 16, 32, 64$ cases here and for $m \geq 16$ cases, we only show the nearest neighbors of the first 16 raw images.

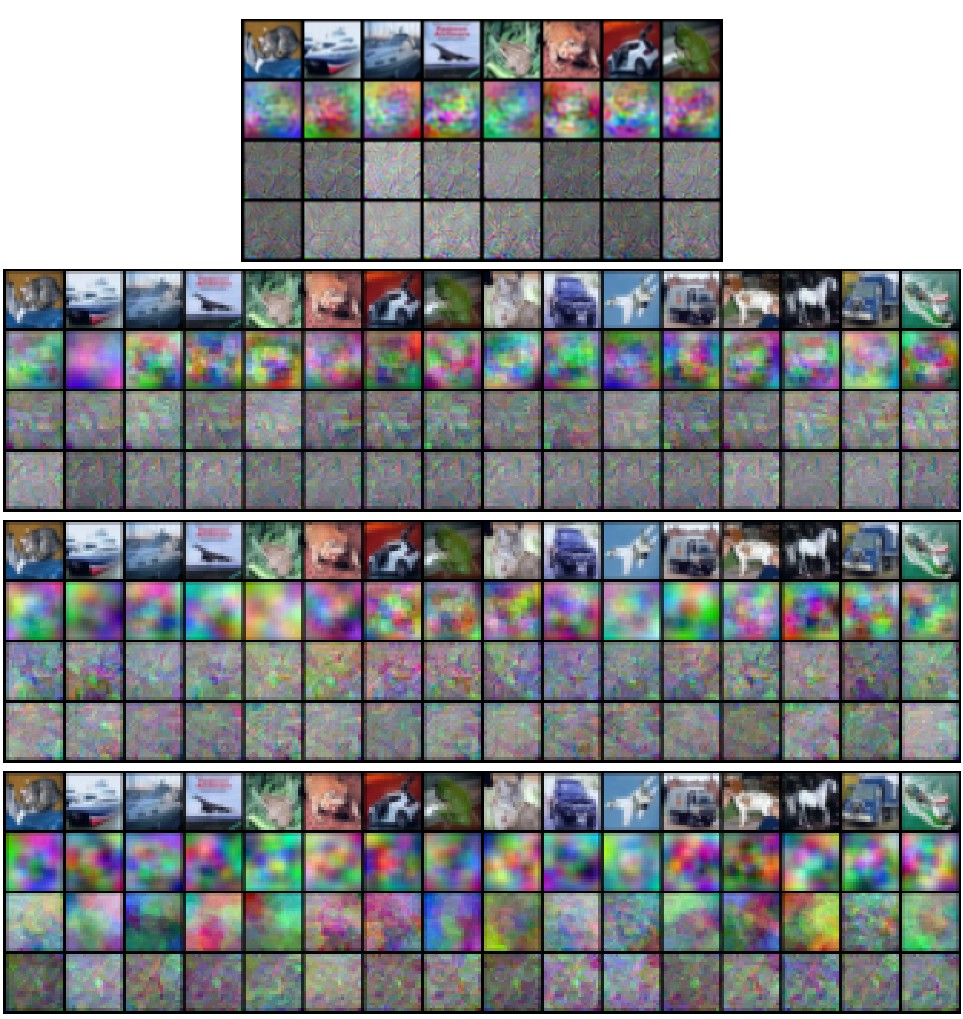

Figure 17: Attacking from features of Scattering net on CIFAR10 dataset. The first row corresponds to raw images, and the following three correspond to recovered images by MIA with $\sigma = 0$, $\epsilon = 8$, and $\epsilon = 1$, respectively. Due to the limited space, we present $m = 8, 16, 32, 64$ cases here, and for $m \geq 16$ cases, we only show the nearest neighbors of the first 16 raw images.

