# OpenReview forum: "NeuroMixGDP: A Neural Collapse-Inspired Random Mixup for Private Data Release"
_CPAL.cc/2024/Conference — CPAL 2024 (Proceedings Track) Oral_

### Official Review · Reviewer_3QmT · 2023-10-07
**Review of NeuroMixGDP**

**Rating:** 6
**Confidence:** 3

**Review:**

Personally, I have found the exposition of this paper quite hard to follow. I do not think it is clearly outlined in the abstract and introduction how the authors leverage insights from the NC phenomenon to improve the utility of differentially private mixup. With that said, here is a summary of my reading.

Summary:

Previously, Zhang et al. proposed RW-Mix, a data augmentation technique aimed at improving the utility of (presumably) ML algorithms. However, when noise is introduced in a differentially private context, the algorithm's performance is compromised, resulting in diminished utility (whether that is for data publishing or for further use). To address this issue, this paper introduces two main algorithms: NeuroMixGDP and NeuroMixGDP-HS. The former is a differentially private version of Avg-Mix, while the latter is a differentially private adaptation of NeuroMixGDP, incorporating hierarchical sampling.

Review:

Overall, I find the idea in this paper to be novel: using insights from label collapse to propose a new algorithm in differentially private data publishing. Furthermore, the results appear to be very promising — the proposed method seems to significantly outperform existing methods, as reported in Table 2. The main issue I have is with the clarity of the paper, which I outline below:
- It is not made clear why Poisson sampling suffers from the Label Collapse issue other than being an empirical result. Is there an intuition for this, and does this apply only to Poisson sampling or more generally? If it's specific to Poisson sampling, can we employ different sampling techniques to avoid label collapse?
- Is the linear model an appropriate choice for studying and characterizing the sweet spot of the mixup degree? Were there other models that could have been considered, and if so, what are their advantages and limitations? The authors do not clearly explain why they opted for this specific model.
- I believe that Section 5, which covers the membership attack, could provide more detail and further motivation on why this is a valuable case study.
- To my knowledge, saying that an algorithm is “\mu-GDP” is not customary. In the relevant literature, a Gaussian differentially private algorithm is generally referred to as (\epsilon, \delta)-DP.

I would be more than willing to raise my score if these questions can be well addressed. As suggested by my confidence score, there may have been some parts that I may have misunderstood.

---

### Official Review · Reviewer_T1Dw · 2023-10-07
**A new differential privacy approach inspired from Neural Collapse**

**Rating:** 7
**Confidence:** 3

**Review:**

This paper aims to mix up the Neural Collapse features to enhance the utility of the privacy-preserving data. The research topic of privacy preserving is crucial for the ML community. The main idea underlying NeuroMixGDP is instead of mixuping input features, the mixture of output features should make more sense, as the former may suffer from the broken 'Non-Approximate Collinearity (NAC)' condition. Their preliminary experiments verify NC indeed induces NAC and provide empirical evidence to support the main claim of this paper. Two technical contributions, Averaging Mixup and Hierarchical Sampling, were proposed to significantly boost the performance of NeuroMixGDP. Experimental results also verify the effectiveness of NeuroMixGDP.

Strengthness:

(1) The paper is well-motivated and supported by results.

(2) I like the preliminary results provided in Figure 1 and Figure 2, which provides clear evidence and important insights to readers.

(3) The authors also provide results of Model Inversion Attack, making the paper more solid.

Weakness:

(1) Can the authors elaborate more the experimental settings of Figure 1 and Figure 2. I would like to see the similar trend on various settings and therefore, the finding in this paper is a general one that can be observed across various settings.

(2) Are we solely extracting the output features of the last layer or all layers? I expect only the features from last layer are helpful here.

(3) What does epsilon in Figure 2 stand for?

(4) I encourage the authors to explain the Poisson Sampling in the early part of the paper.

(5) what is the formulation or definition of sensitivity mean here? what its relationship to DP noise?

---

### Official Review · Reviewer_WntU · 2023-10-07
**see below**

**Rating:** 7
**Confidence:** 3

**Review:**

Summary:

NeuroMixGDP is a novel approach to privacy-preserving data release that proposes a new mixup scheme inspired by the Neural Collapse phenomenon. The paper discusses the challenges that can arise when using the feature mixup framework, such as the sensitivity blowup of RW-Mix and label collapse. The authors examine how Avg-Mix and HS can be used to address these issues, respectively, and how these approaches informed the development of their novel designs, NeuroMixGDP(-HS). The paper provides the asymptotically optimal mixup degree rate using GDP. The proposed method is shown to significantly enhance the utility of released data while protecting user privacy. The paper also discusses the effectiveness of the proposed method in defending against attacks, such as model inversion attack and membership inference attack. Overall, the paper presents a promising approach to privacy-preserving data release that can improve the utility of released data while protecting user privacy.

Pros:

1. Enhances the utility of released data: The proposed mixup scheme can significantly improve the utility of released data, making it more useful for downstream machine learning tasks.
2. Asymptotically optimal mixup degree rate: The paper provides the asymptotically optimal mixup degree rate using basic linear model, which can help to understand the "sweet spot" choice of mixup degree.
3. Defends against attacks: The paper demonstrates that the proposed method can defend against attacks such as model inversion attack and membership inference attack.

Cons:
1.  Lacks some definitions: some core definitions are lack (utility and sensitivity), which poses certain difficulties to readers who are unfamiliar with the DP area.
2.  Requires further validation: while the paper presents promising results, the approach will need to be further validated and tested in larger datasets scenarios to determine its effectiveness and practicality.
3. Lacks neural collapse reference: The related neural collapse references are very minimal. It does not cite many of the theoretical works in NC literature (e.g. [1-7])
4. Some typos: the equation of y in definition 2.1 should use index j rather than i?  "While the ... m samples" in line 131-133 is not a complete sentence.

Reference:

[1] Ji, Wenlong, et al. "An unconstrained layer-peeled perspective on neural collapse." arXiv preprint arXiv:2110.02796 (2021).

[2] Zhu, Zhihui, et al. "A geometric analysis of neural collapse with unconstrained features." Advances in Neural Information Processing Systems 34 (2021): 29820-29834.

[3] Han, X. Y., Vardan Papyan, and David L. Donoho. "Neural collapse under mse loss: Proximity to and dynamics on the central path." arXiv preprint arXiv:2106.02073 (2021).

[4] Zhou, Jinxin, et al. "On the optimization landscape of neural collapse under mse loss: Global optimality with unconstrained features." International Conference on Machine Learning. PMLR, 2022.

[5] Zhou, Jinxin, et al. "Are all losses created equal: A neural collapse perspective." Advances in Neural Information Processing Systems 35 (2022): 31697-31710.

[6] Mixon, Dustin G., Hans Parshall, and Jianzong Pi. "Neural collapse with unconstrained features." arXiv preprint arXiv:2011.11619 (2020).

[7] Tirer, Tom, and Joan Bruna. "Extended unconstrained features model for exploring deep neural collapse." International Conference on Machine Learning. PMLR, 2022.

---

### Meta-Review · Area_Chair_xNCg · 2023-11-14

**Recommendation:** Accept (Poster)
**Confidence:** 4

**Metareview:**

This paper introduces a novel method for private data release, effectively balancing data utility and privacy. The approach, inspired by the neural collapse phenomenon, addresses key challenges in existing privacy-preserving algorithms. The authors have commendably responded to reviewers' feedback, enhancing the paper's clarity and empirical validation. The promising results and novel application of neural collapse insights make this submission a significant contribution to the field. Hence, I recommend its acceptance for CPAL 2024.

---

### Decision · Program_Chairs · 2023-11-19

**Decision:**

Accept (Oral)

**Comment:**

All reviewers and AC agreed that the paper is of high quality. This paper introduces a novel method for private data release, effectively balancing data utility and privacy. The approach, inspired by the neural collapse phenomenon, addresses key challenges in existing privacy-preserving algorithms. The authors have commendably responded to reviewers' feedback, enhancing the paper's clarity and empirical validation. The promising results and novel application of neural collapse insights make this submission a significant contribution to the field.

The action PC chair for this paper is Qing Qu, who made the decision after carefully reading the paper as well as the comments by all reviewers and AC. The decision is agreed upon by all PC chairs.